# *Nrp2* is sufficient to instruct circuit formation of mitral-cells to mediate odour-induced attractive social responses

Kasumi Inokuchi[1,2], Fumiaki Imamura[3], Haruki Takeuchi[1], Ryang Kim[4], Hiroyuki Okuno[4], Hirofumi Nishizumi[1,2], Haruhiko Bito[4], Takefumi Kikusui[5] & Hitoshi Sakano[1]

Odour information induces various innate responses that are critical to the survival of the individual and for the species. An axon guidance molecule, Neuropilin 2 (Nrp2), is known to mediate targeting of olfactory sensory neurons (primary neurons), to the posteroventral main olfactory bulb (PV MOB) in mice. Here we report that Nrp2-positive (Nrp2[+]) mitral cells (MCs, second-order neurons) play crucial roles in transmitting attractive social signals from the PV MOB to the anterior part of medial amygdala (MeA). Semaphorin 3F, a repulsive ligand to Nrp2, regulates both migration of Nrp2[+] MCs to the PV MOB and their axonal projection to the anterior MeA. In the MC-specific Nrp2 knockout mice, circuit formation of Nrp2[+] MCs and odour-induced attractive social responses are impaired. *In utero*, electroporation demonstrates that activation of the *Nrp2* gene in MCs is sufficient to instruct their circuit formation from the PV MOB to the anterior MeA.

[1] Department of Brain Function, School of Medical Science, University of Fukui, Fukui 910-1193, Japan. [2] Department of Biophysics and Biochemistry, Graduate School of Science, The University of Tokyo, Tokyo 113-0032, Japan. [3] Department of Pharmacology, Pennsylvania State University College of Medicine, Hershey, Pennsylvania 17033, USA. [4] Department of Neurochemistry, Graduate School of Medicine, The University of Tokyo, Tokyo 113-0033, Japan. [5] Department of Animal Science and Biotechnology, School of Veterinary Medicine, Azabu University, Sagamihara, Kanagawa 252-5201, Japan. Correspondence and requests for materials should be addressed to H.S. (email: sakano.hts@gmail.com).

The mammalian olfactory system recognizes a diverse repertoire of chemical information that induces distinct behavioural responses based on the odour qualities[1]. In rodents, odour ligands are detected by odorant receptors (ORs) of olfactory sensory neurons (OSNs) in the olfactory epithelium (OE)[2]. Since OSNs expressing the same type of OR send their axons to a specific target site, glomerulus, and each odorant can interact with multiple OR species[3–5], odour signals detected in the OE are converted into a topographic map of activated glomeruli. Odour information encoded in the olfactory bulb (OB) is then conveyed by projection neurons, mitral/tufted (M/T) cells, to various areas of the olfactory cortex (OC) to elicit odour responses[6,7]. How is it then that the map information is transmitted from the OB to the OC for behavioural decisions? For olfactory map formation, OSN axons are guided to approximate locations in the OB by using different sets of axon guidance molecules[8–11] and the map is further refined in an activity-dependent manner[12]. In contrast to the primary projection of OSNs, molecular mechanisms for the circuit formation of M/T cells are largely unknown.

How is the odour map interpreted and how are odour qualities determined for behavioural decisions? The olfactory map is not merely a projection screen that recognizes patterns to discriminate various odorants, but it is composed of functional domains that induce innate odour responses[13]. We have previously reported that the avoidance domain for spoiled foods and the fear domain for predator odours are separately located in the dorsal part of the OB, in $D_I$ and $D_{II}$ regions, respectively[14]. The cortical amygdala (CoA) is reported to mediate odour information of aversive smells[15,16]. As for the attractive social cues, responding glomeruli are found in the posteroventral (PV) part of the OB. For example, male urine compounds methylthio–methanethiol (MTMT)[17] and (Z)-e-tetradecen-1-ol (5Z-14:OH)[18] are detected by MOR83 and MOR286-3P, respectively, whose glomeruli are located in the PV OB[19]. Although neural circuits have not been established for these social cues, the medial amygdala (MeA) is reported to respond to odour inputs for attractive responses[20]. We assume specific circuit links between functional domains in the OB and particular regions in the OC to induce behavioural outputs.

The MeA is known to elicit pheromone-induced innate behaviours by receiving signals from the accessory OB (AOB)[16,20,21]. However, the MeA receives social odour signals also from the main OB (MOB). Dye-injection and pharmacological ablation experiments support this notion[22,23]. Furthermore, some attraction behaviours mediated by the MeA are not affected in the knockout (KO) of TrpC2 (AOB-specific ion channel)[24–26], but impaired in the KO of CNG-A2 (MOB-specific ion channel)[27]. Although these previous experiments indicate that the main olfactory system plays a critical role in inducing attractive social responses, the exact mechanism for how odour signals are correctly transmitted from the MOB to the MeA remains uncertain. It is also unclear whether the second-order neurons, M/T cells, can be divided into different subsets for specific roles and how their cell lineages are determined during development. Thus, it is quite important that the functional and structural connectivity between the MOB and the OC be elucidated.

Here we report that a subset of mitral cells (MCs) that are Neuropilin 2-positive (Nrp2$^+$) play a critical role in transmitting attractive social signals from the PV MOB to the anterior MeA. In the MC-specific conditional KO (cKO) of Nrp2, both circuit formation of Nrp2$^+$ MCs and odour-induced social attraction behaviours are impaired. Furthermore, in utero, electroporation demonstrates that activation of the Nrp2 gene in MCs is sufficient to instruct circuit formation from the PV MOB to the anterior MeA.

## Results

**Nrp2$^+$ MCs in the MOB elicit social attraction responses.** We have previously reported that Nrp2$^+$ OSN axons are guided from the ventral OE to the PV MOB by repulsive interactions with Semaphorin 3F (Sema3F) secreted by dorsal OSN axons[11]. Furthermore, a subpopulation of MCs in the PV MOB are Nrp2$^+$ (refs 11,28). These observations suggested an intriguing possibility that Nrp2$^+$ MCs may be responsible for receiving attractive social signals from the Nrp2$^+$ glomeruli and transmitting the signals to the anterior MeA. We, thus, examined whether odour-induced social behaviours are affected in the Nrp2 cKO. MC-specific cKO mice were generated by crossing the Nrp2-floxed mice[29] with the AP2ε (immature MC marker)-Cre as a driver[30,31]. AP2ε has restricted expression in the OB and represents the earliest known marker for the developing OB[30–32]. To demonstrate the tissue-specific activity of Cre, we crossed the AP2ε-Cre line to the ROSA-stop-lacZ mouse[30]. Heavy X-gal staining was found in the MC layer in the OB (Supplementary Fig. 1a); however, OSNs in the OE were devoid of staining.

Since the PV MOB is reported to be activated by urinary volatiles[23,33], we evaluated the response of male mice to female urine that was physically separated from the male mice so that only volatile social cues could be detected (Fig. 1a, left). In the control experiment, the heterozygous littermate male was strongly attracted to the female urine, sniffing it for much longer period of time compared to that of another male. In the male cKO of Nrp2, no difference was observed in sniffing between the female and male urine (Fig. 1a, right). We also examined whether the MTMT-induced female behaviours[17] are affected in our cKO of Nrp2. It was found that investigation times of female mice were significantly lowered in the cKO towards MTMT with castrated male urine (Fig. 1b). Interestingly, female cKO demonstrated increased attraction to the castrated male urine alone compared with controls, which may be explained by alterations of neural circuitries in the Nrp2 cKO. These results supported the idea that Nrp2$^+$ MOB MCs are involved in eliciting social attraction behaviours. However, one could not exclude the possibility that some diffusible urine components might have been processed by the vomeronasal organ (VNO). We, therefore, studied male-mouse ultrasonic vocalization (USV)[34] that is regulated independently from the VNO[24]. USV signals emitted by resident males were compared between the cKO and heterozygous littermates in the presence of a female intruder. It was found that USV by males towards females was diminished in the MC-specific cKO of Nrp2 (Fig. 1c,d and Supplementary Fig. 1b).

We next studied suckling behaviour of pups in the MC-specific cKO of Nrp2. It is known that in neonatal rabbits, nipple search is a stereotyped innate behaviour regulated by maternal odour substances and independent of the AOB[35]. Since the VNO is not fully functional until postnatal day 10 (P10) in mice[36], it is unlikely that suckling behaviour of pups is regulated by the VNO, but instead by the main olfactory system. In the control pups (heterozygous littermates), ∼90% were able to successfully find their mother's nipples within 4 min (Fig. 1e). In contrast, <20% of mutant pups were able to locate the nipples. However, this difference was not due to any defect in locomotive activities. We placed pups on their back and determined the time they spent to recover from this position (Supplementary Fig. 1c). In this reflex righting test, no difference was found between the cKO and control littermates.

We previously reported that innate fear responses induced by a fox odour, trimethylthiazoline (TMT), are mediated by the dorsal-region glomeruli in the MOB[14]. Furthermore, the CoA is known to receive aversive information of TMT[15,16]. To determine

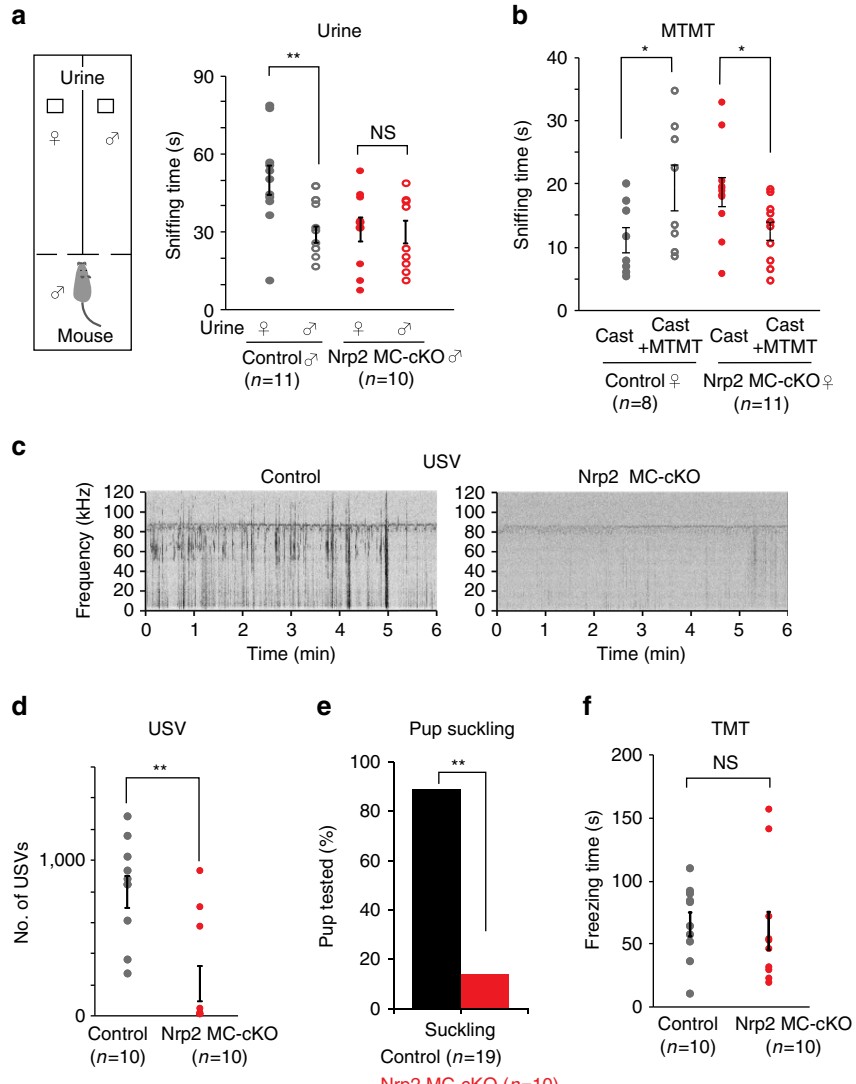

**Figure 1 | Innate social behaviours in the MC-specific cKO of Nrp2.** (**a**) The male attractive behaviour towards female urine. A piece of filter paper spotted with 60 µl of male or female urine was presented to the male cKO or heterozygous littermate as a control. Sniffing time duration was measured within 5 min after presenting the filter paper. Although the control male spent longer times sniffing female urine, the cKO male did not show any preference. Data are presented in mean ± s.e. **$P < 0.01$, NS: not significant (Student's $t$-test). (**b**) The female attractive behaviour towards MTMT. A piece of filter paper spotted with 50 µl of castrated male urine or castrated male urine with 20 p.p.b. MTMT was presented to the female cKO. Sniffing time duration was measured within 5 min after presenting the filter paper. Data are presented in mean ± s.e. *$P < 0.05$ (Student's $t$-test). (**c**) Spectrograms of USVs. USV signals emitted by resident males are compared between the cKO and heterozygous littermates in the presence of a female intruder. The intruder female was introduced into the cage for 6 min in each trial. Very few signals are detected in the cKO male. (**d**) Comparison of numbers of USV signals elicited by male mice in the presence of a WT female. USV is significantly reduced in the cKO. Data are presented in mean ± s.e. **$P < 0.05$ (Wilcoxon signed-rank test). (**e**) Suckling behaviour in the Nrp2 MC-cKO pups. Lactating female mice were anaesthetized and laid sideways, exposing their nipples towards pups. Pups were fasted for 4 h before the experiment and placed ∼1 cm away from the females' nipples. The time to find nipples for suckling was measured within 4 min. Most cKO pups failed to find nipples within 4 min. ($n = 19$ for the control, $n = 10$ for the MC-specific cKO of Nrp2). **$P < 0.01$ (Z-test). (**f**) Fear responses to a predator's odour. A piece of filter paper spotted with 10 µl of undiluted TMT was presented to the cKO and heterozygous littermate as a control. Freezing time lengths were measured during the 10 min exposure to TMT. NS: not significant (paired $t$-test). Data are presented in mean ± s.e.

if $Nrp2^+$ MCs are involved in eliciting fear responses to TMT, we examined TMT responsiveness in the MC-specific cKO of Nrp2. Interestingly, TMT-induced fear responses were not affected in the cKO (Fig. 1f). These observations demonstrate that the absence of Nrp2 in MOB MCs interferes with social attraction behaviours, while having no effect on the TMT-induced innate fear responses that are mediated by the dorsal MOB. Furthermore, no abnormality was found in the detection of odour substances, such as α-terpinene and vanillin, that activate the dorsal glomeruli[37,38]. No difference was found in the

habituation–dishabituation test between the cKO and heterozygous littermate mice (Supplementary Fig. 1d, left) even with the $10^{-5}$ dilution of vanillin for dishabituation (Supplementary Fig. 1d, right).

**$Nrp2^+$ MCs are guided to the PV MOB by Sema3F.** Having demonstrated that Nrp2 is essential for critical social behaviours, such as USV communication, we could directly test the hypothesis that Nrp2 is involved, and potentially instructive, in neural

circuit formation between the MOB and the anterior MeA. The significant changes in odour-induced social behaviours can be explained by the alterations in the neural circuitry in the Nrp2 cKO. To examine this, we first analysed migration and distribution of Nrp2$^+$ MCs in the MOB of the wild-type (WT) mice. In the embryo, MC precursors are generated in the ventricular zone at E10–13 and migrate radially, through the intermediate zone, to the mitral cell layer (MCL) in the MOB[39,40] (Supplementary Fig. 2a). Double in situ hybridization, using Pcdh21 (mature MC marker) and Nrp2 probes, revealed that the Nrp2 gene is expressed in a subset of MCs in the WT (Fig. 2a). Nrp2 expression gradually decreases after birth and disappears by PD14. We, therefore, attempted to analyse how the two types of MCs, Nrp2$^+$ and Nrp2$^-$, are generated in the MOB during development. Pcdh21 and AP2ε were used as developmental markers for mature and immature MCs, respectively. At E13.5, immature MCs (AP2ε$^+$) are located within the intermediate zone, where the Nrp2$^+$ and Nrp2$^-$ cells are already separated (Supplementary Fig. 2b). In situ hybridization revealed that not all MCs in the PV MOB are Nrp2$^+$ (Supplementary Fig. 2b). There may be another subset of MCs in the PV MOB, which is Nrp2$^-$. It is possible that these Nrp2$^-$ MCs were originally Nrp2$^+$, but lost Nrp2 expression after migrating to the PV. Our BrdU experiment indicates that both Nrp2$^+$ and Nrp2$^-$ MCs are simultaneously produced at E10–13 (Supplementary Fig. 2c). At P0, when most MCs have reached the MCL, Nrp2$^+$ MCs are confined to the PV region of the MOB (Fig. 2a). As for the

primary neurons, dorsal OSN axons are Nrp2$^-$, whereas the PV OSN axons are Nrp2$^+$ (Fig. 2b)[11]. Since MCs PV tend to extend their primary dendrites perpendicularly to the nearest glomerulus, Nrp2$^+$ MC dendrites make synaptic connections with the Nrp2$^+$ OSN axons in the PV MOB. This dorsal/ventral partitioning appears to provide a topographical and functional separation of the olfactory map[11,14].

To specifically examine how Nrp2$^+$ MCs are guided to their proper locations in the MCL, and whether Nrp2 regulates the association between OSN axons and MC dendrites, we analysed the Nrp2 KO mouse[29]. In this total KO, a coding sequence of Nrp2 was replaced with that of GFP, so that cells with activated Nrp2 promoter could be detected with the GFP probe. Our studies showed that not only the targeting of OSN axons (Fig. 3a, upper panel and Supplementary Fig. 3a), but also the distribution of MCs is affected in the KO mice (Fig. 3a, lower panel and Supplementary Fig. 3b). In contrast to the total KO, the MC-specific cKO of Nrp2 did not affect the targeting of Nrp2$^+$ OSN axons. Likewise, MC migration was not affected in the OSN-specific cKO of Nrp2 (Fig. 3a and Supplementary Fig. 3b). On the basis of these observations, the possibility of direct crosstalk between the OSN axons and MC dendrites via Nrp2 molecules was excluded. We, therefore, focused our attention on Sema3F, a repulsive ligand for Nrp2. Sema3F is secreted by the dorsal OSN axons to repel the ventral OSN axons and is not produced by MCs in the MOB[11]. To examine whether the OSN-derived Sema3F also regulates the migration of MCs, we analysed the distribution of Nrp2$^+$ MCs in the OSN-specific cKO of Sema3F. In this cKO, distribution of Nrp2$^+$ MCs in the MOB is severely affected: Nrp2$^+$ MCs are found not only in the ventral, but also in the dorsal part of the MCL (Fig. 3b, lower panel and Supplementary Fig. 3d). In the same Sema3F cKO, Nrp2$^+$ OSN axons are also defasciculated in the glomerular layer (GL) (Fig. 3b, upper panel and Supplementary Fig. 3c). These phenotypes are similar to those observed in the total KO of either Sema3F or Nrp2 (Fig. 3c,d). We performed these targeting experiments at both E18 and P0 and obtained basically the same results. However, signals were clearer at E18 than P0 in the hybridization experiment probably due to the differences in transcription levels of Nrp2. In contrast, we obtained better staining at P0 than E18 in the immunohistochemistry. Taken altogether, we conclude that for proper matching to occur, OSN-derived Sema3F in the dorsal MOB regulates both targeting of Nrp2$^+$ OSN axons and migration of Nrp2$^+$ MCs to the PV MOB.

**Nrp2$^+$ MCs target their axons to the anterior MeA.** To study a specific function of Nrp2$^+$ MCs in inducing social attraction behaviours, we next analysed their axonal projection and arborization in the OC. MCs send their axons to various brain regions by following the lateral olfactory tract[6,41–44] (Fig. 4a, right). Since Nrp2 expression was not sufficiently strong for tracing axons by immunostaining, we developed a transgenic (Tg) mouse in which the fluorescent reporter protein, EYFP, was expressed in Nrp2$^+$ MCs. To do this, we generated Tg mice carrying the transgene, pcdh21-lacZ-STOP-tauEYFP, in which the Cre-inducible tauEYFP gene was expressed with the Pcdh21 promoter (Fig. 4a). The lacZ gene was introduced into the construct to monitor the Pcdh21 promoter activity (Supplementary Fig. 4a, left). Four independent mouse lines, that successfully expressed the lacZ gene in MCs, were obtained. While the Pcdh21 promoter is usually active in both the MOB and the AOB, in one of our four mouse lines, the transgene was activated only in the MOB and not in the AOB (Supplementary Fig. 4a, right). Signals of β-gal seen around the AOB are MC axons passing by from the MOB. This mouse line proved quite useful for analyzing MOB MCs for

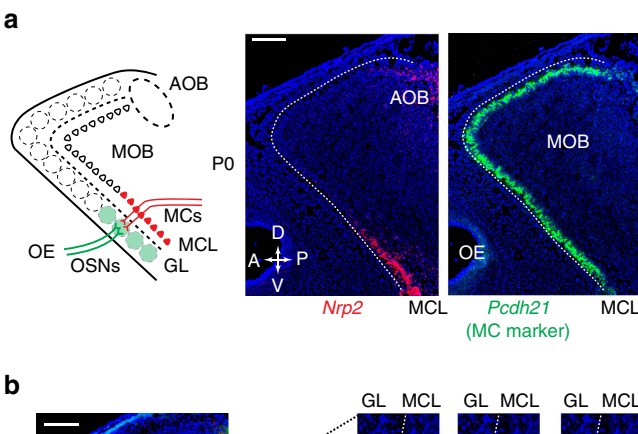

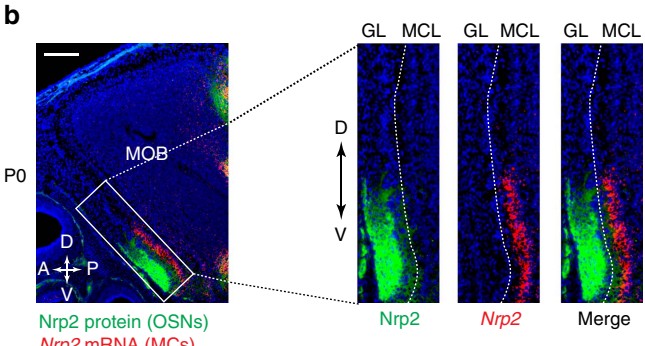

**Figure 2 | Distribution of Nrp2 in the GL and MCL of the MOB.**
(**a**) Detection of Nrp2 transcripts in the ventral-region MCs. Parasagittal MOB sections (P0) were hybridized with Pcdh21 (MC marker) and Nrp2 probes. n = 5. Nrp2 is expressed in the MCL, high in the ventral and low in the dorsal region. Nrp2$^+$ signals seen in the PD region represent Nrp2$^+$ MCs in the AOB. A schematic diagram of the mouse MOB is shown in the left. (**b**) Distribution of Nrp2 protein and Nrp2 transcripts in the MOB. MOB sections at P0 were immunostained with anti-Nrp2 antibodies or hybridized with the Nrp2 probe. Nrp2 and Nrp2 signals are distributed in parallel along the D–V axis in the GL and MCL, respectively. Enlarged photos are shown in the right. n = 5. Scale bar, 200 μm.

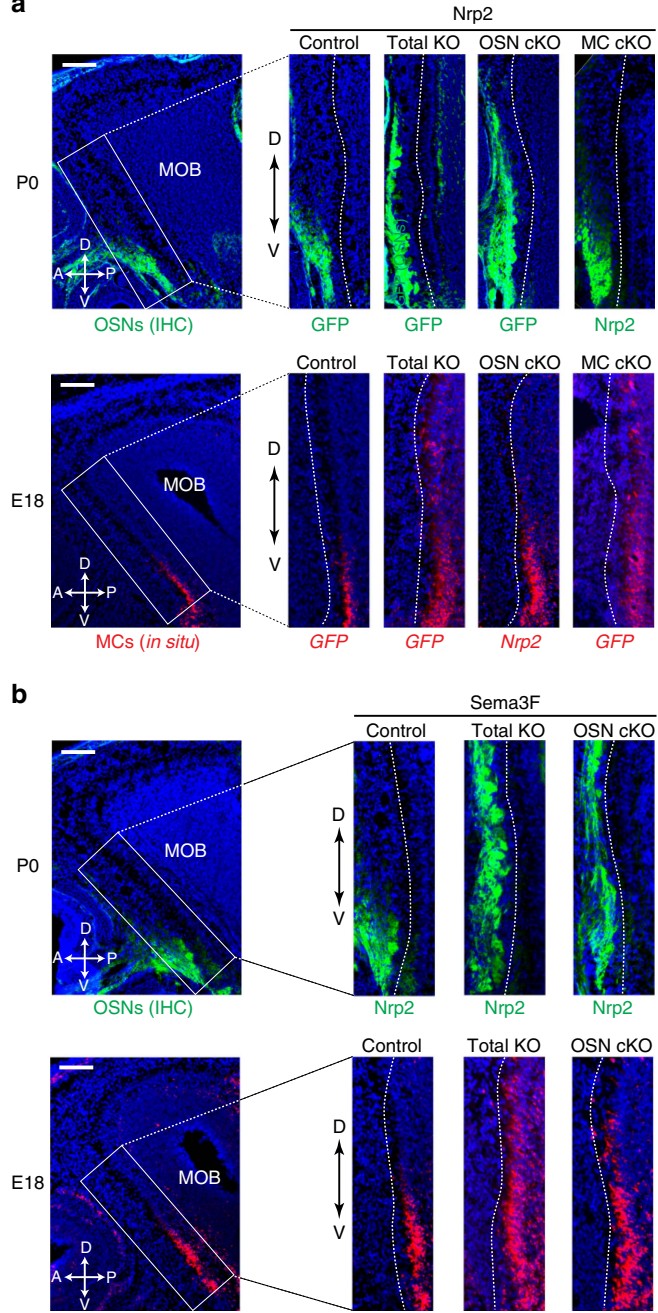

detected in the *Nos1*[+] MeA (Supplementary Fig. 4c), but are negative for OCAM, a marker for dorsal-region MCs[44] (Fig. 4b, right). Here we analysed the cortical projection at P14, because MC targeting is completed at P14. However, OCAM staining was performed at P0, because OCAM expression decreases after birth. We also observed that the EYFP[+] region is positive for Nrp1, a marker for MOB MCs (Supplementary Fig. 4d, left). Double ISH confirmed that *Nrp1* is expressed in the Nrp2[+] MCs (Supplementary Fig. 4d, right). These results clearly demonstrate that Nrp2[+] MCs in the MOB connect the PV-region MOB and anterior MeA.

**Trans-synaptic labelling of MCs in the MOB.** To examine whether anterior MeA neurons selectively receive afferent projections from the PV MOB, we performed monosynaptic labelling with the recombinant rabies virus (RV), which transmits across synapses in a retrograde direction[48]. For the labelling of MCs, we used recombinant virus RV-ΔG-EGFP-EnvA that requires target cells to express avian receptor protein TVA and glycoprotein G for its selective infection, as well as monosynaptic transmission (Fig. 5a). We first injected adeno-associated virus (AAV) into the anterior side of MeA to complement TVA and glycoprotein G (Fig. 5b, left and Supplementary Fig. 5). After 2 weeks, RV-ΔG-EGFP-EnvA was injected into the same MeA region to selectively infect neurons expressing the TVA receptor. After 10 more days, serial sections of the entire MeA were analysed to confirm that the neurons infected with the AAV and RV were indeed confined to the MeA (Fig. 5b, left). We then examined the locations of EGFP-labelled MCs that were retrogradely infected with the RV. Eighty-one starter cells gave rise to 179 labelled MCs, 150 of which were specifically found in the PV MOB (Fig. 5b, right), suggesting that the MeA receives biased inputs from the PV MOB. For a control experiment, we injected the AAV and RV into the PIR, where 478 labelled MCs were generated from 98 starter cells in the PIR (Fig. 5c, left). No special preference was found in their locations in the MOB (Fig. 5c, right): EGFP-positive MCs are distributed randomly in the MOB as reported[49]. These observations further support the idea that the anterior MeA receives MC projection from the MOB from the PV region.

**Targeting of Nrp2[+] MCs is also regulated by Sema3F.** How do the Nrp2[+] MCs project their axons from the PV MOB to the anterior MeA? To examine the possibility that Nrp2 expressed by MCs plays a critical role in guiding axons, the MC-specific cKO of Nrp2 was analysed at P14 for MC projection from the MOB. Since Nrp2 signals are hardly detected at this stage, anti-Nrp1

**Figure 3 | Sema3F co-regulates OSN projection and MC migration in the MOB. (a)** Targeting of OSN axons and MC migration in the Nrp2 KOs. Parasagittal MOB sections at P0 were analysed by immunohistochemistry (upper panel). In the KOs, a coding sequence of the *Nrp2* has been deleted and replaced with that of *GFP*. Thus, OSNs that are supposed to express Nrp2 can be detected with anti-GFP antibodies. Parasagittal sections at E18 were also analysed by *in situ* hybridization using the *GFP* and *Nrp2* probes (lower panel). Dotted lines in the enlarged photos indicate the boundary of the GL and MCL. *n* = 7, 10, 6 and 8 for the control, total KO, OSN-specific cKO and MC-specific cKO, respectively. **(b)** Targeting of OSN axons and MC migration in the Sema3F KOs. Parasagittal MOB sections at P0 were immunostained with anti-Nrp2 antibodies (upper panel). In the Sema3F total KO and OSN-specific Sema3F conditional KO (OSN cKO), Nrp2[+] OSNs are defasciculated and mistargeted to the dorsal MOB. Dotted lines in the enlarged photos indicate the boundary of the GL and MCL. Parasagittal MOB sections at E18 were also analysed by *in situ* hybridization with the *Nrp2* probe (lower panel). *n* = 7, 9 and 6 for the control, total KO and OSN-specific cKO, respectively. Scale bar, 200 μm. D, dorsal; V, ventral; A, anterior; P, posterior.

axonal projection separately from AOB counterparts. To activate the *EYFP* gene specifically in the Nrp2[+] MCs of the MOB, the mouse was then crossed with another Tg mouse containing the *Nrp2-Cre*, where Cre-recombinase was induced by the *Nrp2* promoter (Fig. 4a). As expected, EYFP signals were detected only in the PV-region MCs, but not in the dorsal ones (Supplementary Fig. 4b). This Tg mouse also allowed us to trace the trajectory of axonal projection of Nrp2[+] MCs by immunostaining EYFP.

MCs in the MOB have been reported to project their axons to various regions of the OC, including the anterior olfactory nuclei (AON), piriform cortex (PIR) and CoA[43,45–47]. Consistent with these previous reports, EYFP[+] axons are found in the superficial layer of these nuclei (Fig. 4b, left). Staining signals of EYFP are uniformly distributed throughout these areas and no clustering of staining signals is observed. Interestingly, EYFP[+] axons are

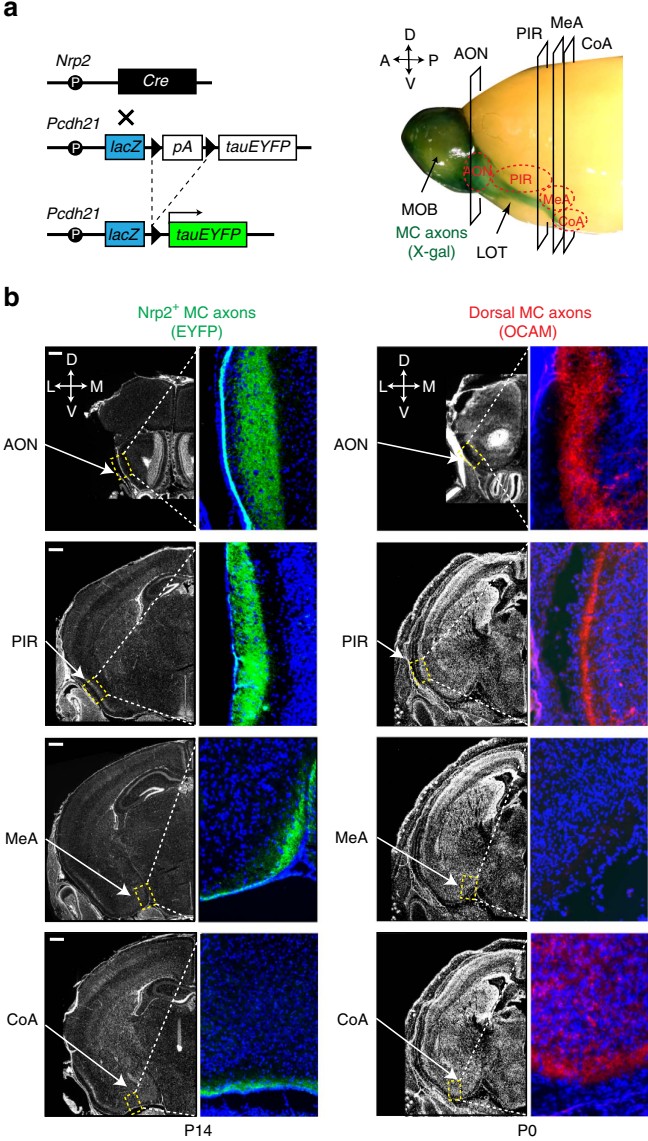

**Figure 4 | Axonal projection of Nrp2⁺ MCs to the MeA.** (**a**) DNA constructs to detect the Nrp2⁺ MC axons (left). The BAC Tg mouse Nrp2-Cre was crossed with another Tg mouse Pcdh21-lacZ-STOP-tauYFP. Since Nrp2 was difficult to detect in MOB MCs, Tg mice, in which the Nrp2-expressing MCs were labelled with EYFP, were generated. The *lacZ* was inserted into the construct to detect the promoter activity of *Pcdh21* by X-gal staining. Whole mount staining of the MOB is shown (right). Locations of coronal sections (AON, PIR and MeA) analysed in **b** are indicated. (**b**) Detection of Nrp2⁺ MC axons in the OC. Coronal sections of the OC at P14 were immunostained with anti-GFP antibodies to detect EYFP expressed in the Nrp2⁺ MCs (left). Nrp2⁺ ventral MC axons (green) project to the AON, PIR, MeA and CoA. Coronal OC sections at P0 were immunostained with anti-OCAM antibodies (right). OCAM⁺ dorsal MC axons (red) project to the AON and PIR, but not to the MeA. n = 6. Scale bar, 100 μm. Nuclei are counterstained with DAPI in blue.

antibodies were used to detect MOB MC axons. Staining signals of Nrp1, a marker for MOB MCs, are severely reduced in the MeA of the Nrp2 KO, although signal levels are not changed in other OC regions (Fig. 6a and Supplementary Fig. 6). This observation demonstrates that the lack of Nrp2 leads to the dramatic decrease in the number of MC axons projecting from the MOB to the anterior MeA (Fig. 9a,c). A similar reduction is also found in the total KO of Sema3F, a repulsive ligand for Nrp2

(Fig. 6a,b). It is interesting that the axonal projection of Nrp2⁺ MCs is not affected by the OSN-specific cKO of Sema3F (Fig. 6a,b and Supplementary Fig. 7a), where the Nrp2⁺ and Nrp2⁻ MCs are not properly segregated within the MOB. This observation suggests that axonal projection of MCs to the anterior MeA takes place independent of their locations in the MOB (Fig. 9b). In the Sema3F total KO, MCs in the AOB also fail to converge to the MeA and mistarget to the CoA (Supplementary Fig. 7b). Since Sema3F is detected in the cortical regions surrounding the MeA (Fig. 7), Nrp2⁺ MC axons are likely guided to the MeA by repulsive interactions with Sema3F in the embryonic OC. Taken altogether, our results indicate that Nrp2 plays a dual instructive role in regulating MC migration within the MOB and in guiding axons of PV-region MCs to the anterior MeA through repulsive interactions with Sema3F expressed in the dorsal MOB and outside regions of the MeA, respectively (Fig. 9a).

***In utero* electroporation of the human *Nrp2* gene.** Although we could demonstrate an essential role of Nrp2 in linking the PV MOB and anterior MeA, it was not clear whether the activation of *Nrp2* is sufficient or any additional genes need to be activated in MCs to induce the circuit formation. To examine how much is regulated by Nrp2, we performed gain-of-function experiments by using *in utero* electroporation. A plasmid vector containing *EGFP* cDNA was introduced into the WT embryonic OB with or without the human *Nrp2* (*hNrp2*) cDNA. As reported previously[50], when the *EGFP* was introduced alone at E11, immunostaining signals for EGFP were equally distributed in both the dorsal and ventral regions of the MCL at P0 (Fig. 8a). In contrast, when the *hNrp2* gene was co-transfected with *EGFP*, EGFP⁺ cells were confined to the PV region in the MOB, as observed in MCs expressing endogenous Nrp2 (Fig. 8a). EGFP⁺ cells were mostly positive for MC markers, Tbr1, Tbr2 and Tbx21 (Fig. 8a and Supplementary Fig. 8). This observation shows that constitutive expression of the exogenous *Nrp2* gene is sufficient to bring MOB MCs to the PV region, even those that would normally remain in the dorsal MOB.

We then analysed axonal projections of MCs in the electroporated WT mice (Fig. 8b). EGFP signals were not strongly found in the MeA when the *EGFP* was electroporated without *hNrp2* (Fig. 8b, upper left). In contrast, EGFP signals were significantly increased in the anterior MeA at P0, when the *hNrp2* was co-transfected with *EGFP* (Fig. 8b, upper right). These experiments indicate that ectopic expression of *hNrp2* can promote not only the migration of MCs to the PV MOB, but also their axonal projection to the anterior MeA. It is interesting that staining signals for OCAM, a dorsal MOB MC marker, were detected in the anterior MeA on the ipsilateral side of the MOB electroporated with *hNrp2* cDNA (Fig. 8b, lower panel). This indicates that constitutive expression of exogenous Nrp2 alone can instruct axonal projection to the anterior MeA even in the dorsal lineage (OCAM⁺) MCs that do not normally send their axons to the anterior MeA. Taken together, these observations clearly demonstrate that activation of *Nrp2* is not only necessary, but also sufficient to instruct circuit formation of MCs from the PV MOB to the anterior MeA.

## Discussion

In the present study, we analysed roles of Nrp2⁺ MCs in generating the neural circuit for attractive social responses. We previously reported that Sema3F secreted by Nrp2⁻ dorsal OSN axons directs late-arriving Nrp2⁺ OSN axons to the PV MOB[11]. The same dorsal OSN-derived Sema3F was found to guide Nrp2⁺ MCs from the ventricular zone to the PV region of MOB. This coordinated regulation of OSN projection and MC

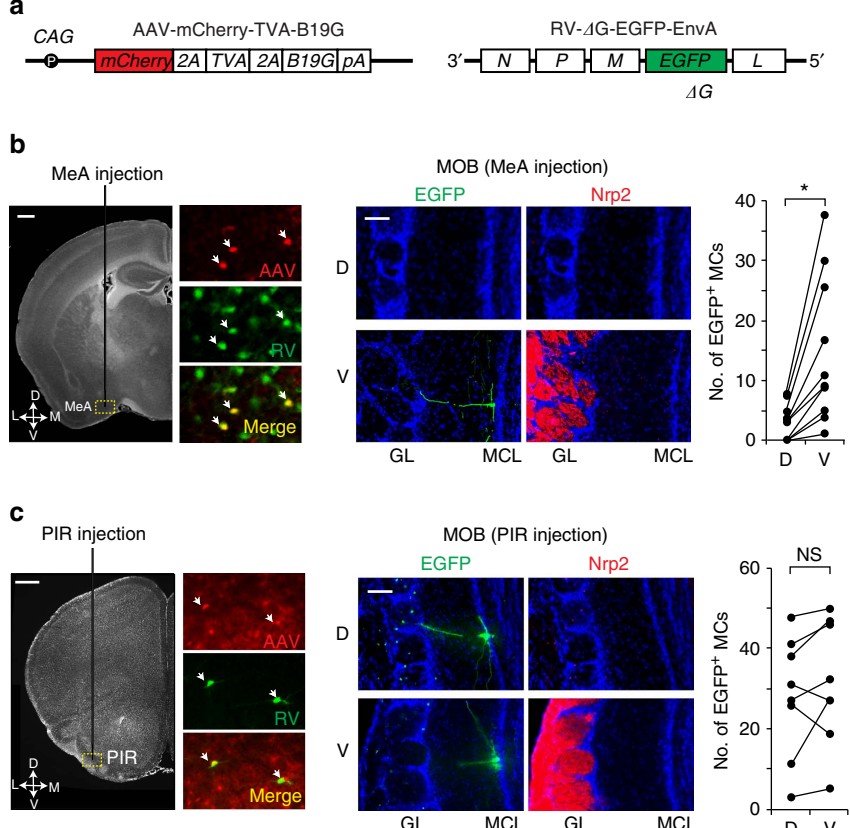

**Figure 5 | Trans-synaptic tracing with the RV.** (a) Virus constructs for circuit tracing. Structures of the AAV and RV are schematically shown. In the RV construct, the G region has been deleted ($\Delta G$) and replaced with *EGFP*[51]. (b) Virus injection into the MeA. AAV was first injected into the MeA region to complement TVA and glycoprotein G. After 2 weeks, the RV was injected into the same MeA region to selectively infect neurons expressing the TVA receptor. After 10 days, serial coronal sections of the entire MeA were analysed to determine whether the neurons infected with AAV and RV were indeed confined to the MeA. Double-labelled cells with mCherry for AAV and EGFP for RV are detected in the MeA. To detect the trans-synaptically labelled MCs, coronal sections of the MOB were immunostained with anti-GFP and anti-Nrp2 antibodies (right). EGFP-positive MCs are detected in the ventral (V), but not in the dorsal (D) region of the MOB. Numbers of EGFP-positive MCs in the dorsal (D) and ventral (V) MCs are compared in the right. Eighty-one starter MeA cells gave rise to 179 labelled MCs ($n = 10$ mice). **$P < 0.05$ (Wilcoxon signed-rank test). Scale bars, 200 $\mu$m (left) and 100 $\mu$m (right). (c) Virus injection to the PIR. Double-labelled cells with mCherry for AAV and EGFP for RV are detected in the PIR (left). To detect the trans-synaptically labelled MCs, coronal sections were immunostained with anti-GFP and anti-Nrp2 antibodies (right). EGFP-positive MCs are detected in both D and V regions of the MOB. Ninety-eight starter cells in the PIR resulted in 478 labelled MCs. $n = 8$ mice. NS: not significant (Wilcoxon signed-rank test). Scale bars, 200 $\mu$m (left) and 100 $\mu$m (right).

distribution appears to be key for functional pairing of OSN axons and MC dendrites in the MOB. We further examined targeting of $Nrp2^+$ MCs localized to the PV MOB. In a previous report[51], dye-injection experiments indicated that the MeA receives MC inputs not only from the AOB, but also from the MOB. By using fluorescent tags, we found that $Nrp2^+$ MCs send their axons to the anterior portion of MeA. Furthermore, retrograde virus tracing revealed that $Nrp2^+$ MCs in the PV MOB do indeed form synapses with MeA neurons.

Our gain-of-function (*in utero* electroporation) and loss-of-function (KO) experiments demonstrate that Nrp2–Sema3F repulsive interactions are essential not only for guiding $Nrp2^+$ MCs to the PV MOB, but also for instructing their axons to the anterior MeA (Fig. 9). In the *in utero* electroporation experiment, MCs expressing hNrp2 with EGFP are not only guided to the PV MOB, but also instructed to send their axons to the anterior MeA. In the electroporated mice, the $EGFP^+$ ($hNrp2^+$) region in the anterior MeA became positive for the dorsal MC marker, OCAM. These results demonstrate that ectopic expression of hNrp2 in the mouse MCs is sufficient to instruct their circuit formation to the anterior MeA even in dorsal lineage MCs. It will be interesting to

study whether these transfected neurons form functional circuits receiving OSN input from the OB and making synaptic connections to the anterior MeA.

In the Nrp2 total KO and MC-specific cKO, signals of MC axons projecting from the MOB are significantly reduced in the anterior MeA, and a similar phenotype is observed in the Sema3F total KO. MCs that were supposed to project to the MeA may reroute their axons targeting to other areas in the OC or simply do not survive during development. Since Sema3F is expressed in the cortical regions surrounding the MeA during early embryonic development, $Nrp2^+$ MC axons are likely driven to the MeA by repulsive interactions with Sema3F expressed just outside their trajectory. It seemed puzzling that $Nrp2^+$ MCs targeted their collaterals to the $Sema3F^+$ areas in the OC at later stages of development. However, this can be explained by the temporal regulation of *Nrp2* and *Sema3F* expression: Both *Nrp2* and *Sema3F* are expressed at embryonic stages (E12–18), but disappear later on when collateral extension starts to occur. It has been reported that the D–V topography in the OB is roughly correlated with that in the AON pars externa but not in the PIR[49,52–54]. Furthermore, incoming odour signals in the CoA are

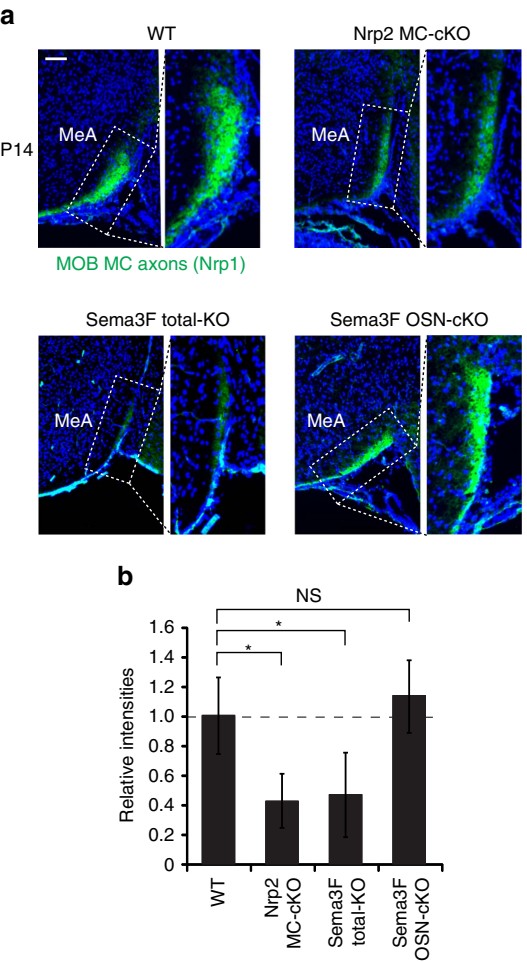

**Figure 6 | MC projection to the MeA is regulated by the Nrp2/Sema3F signalling.** (**a**) Detection of Nrp2[+] MC axons in the MeA. MC axons projecting from the MOB at P14 are stained green with anti-Nrp1 antibodies. Since Nrp2 signals in MOB MCs are hardly detected at this stage, anti-Nrp1 antibodies were used to detect MOB MC axons. Targeting of Nrp1[+] MC axons to the MeA is affected in the MC-specific cKO of Nrp2 and Sema3F total KO, but not in the OSN-specific Sema3F cKO (OSN cKO). Scale bar, 200 μm. (**b**) Nrp1 signals in the MeA. Intensities of green staining (Nrp1[+]) in the anterior MeA are compared between the WT and various KOs. Nrp1 signals are reduced nearly by half in the MC-specific cKO of Nrp2 or Sema3F, but not in the OSN-specific cKO of Sema3F. *$P<0.01$, NS: not significant (Student's $t$-test). $n=5$ for each genotype.

mostly from the dorsal OB[49,53]. Keeping these previous observations in mind, we closely analysed the secondary projection of Nrp2[+] MCs to the AON, PIR, MeA and CoA. We found that Nrp2[+] MCs in the PV OB preferentially target their axons to the anterior MeA, although they send their axons to the AON, PIR and CoA as well.

MCs exhibit a protracted waiting period before sending their collateral branches to the target. It has been reported that the target OC in culture becomes accessible to MC fibres at E16.5 during development[55]. Innervation in the central areas is completed by PD15. First collaterals are extended towards the AON and PIR at E16.5, and innervation to the OT is completed after birth[55,56]. Despite these previous studies, timing of innervation to the MeA was still unclear. Our study demonstrates that Nrp2 instructs targeting of ventral MCs to the MeA. During development, Nrp2 is expressed in the ventral MCs at embryonic stages, but disappears by PD14. It is likely that innervation to the MeA occurs at embryonic stages.

The MeA is known to process non-volatile pheromone signals, detected by the VNO, to induce innate social behaviours, such as mating, conspecific recognition and fear responses[16,20,21,57,58]. Previous dye-injection experiments suggested that the MeA also receive MOB inputs in the anterior regions[51]. Furthermore, a surgical lesion to the anterior MeA of the hamster has been shown to abolish the males' preference for female urine over that of males[16]. Our present study demonstrates that it is the Nrp2[+] subset of MCs that sends attractive social inputs from the PV MOB to the anterior MeA. In the MC-specific cKO of Nrp2, AOB-independent social behaviours are affected. For example, pup suckling and male vocalization are impaired in the cKO. It is generally accepted that the AOB detects non-volatile social cues to induce various innate behaviours, whereas the MOB detects volatile odour ligands[59]. During evolution, the mammalian olfactory system rapidly expanded the MOB structure to accommodate greater numbers of glomeruli for class II ORs devoted to volatile odorants in the ventral region[60]. It is probable that the MOB developed a subset of Nrp2[+] MCs that allowed axons to project to the MeA using the Nrp2/Sema3F guidance system, similar to the AOB. This may have conferred the evolutionary advantage of detecting volatile social cues from distances. It is quite possible that the main olfactory system governs attractive social responses, even in the human whose VNO no longer has any function and vomeronasal-receptor genes have all become pseudogenes[61,62].

It is rather surprising that activation of a single axon guidance gene, *Nrp2*, in immature MCs is sufficient to instruct their neural circuit formation to the anterior MeA. Our present study clearly demonstrates that *Nrp2*[+] MCs in the PV MOB play a critical role in inducing attractive social responses. These findings demonstrate a close correlation between functional domains in the MOB and specific regions in the OC (Fig. 9d). We assume that this is accomplished by gathering glomeruli with the same behavioural quality to a restricted area (functional domain) in the MOB. It is also important for MCs to ensure that the olfactory inputs with the same odour quality, regardless of their chemical nature, are transmitted to a specific region in the OC so that they could induce the same type of outputs. For attractive social responses, Nrp2 is a key determinant for connecting the PV MOB and anterior MeA by using a specific subset of MCs.

It is quite possible that this kind of circuit formation observed in MCs, which is directed by a distinct set of guidance molecules, is a general rule that is applicable to other innate odour responses. For example, fear to predators' smell and avoidance to spoiled food odours are induced by activation of glomeruli located in the avoidance (D_I) and fear (D_II) domains in the MOB, respectively[14]. Furthermore, the CoA is reported to play an important role in inducing aversive responses to fox odour TMT[15]. It will be interesting to determine which axon guidance system is responsible for establishing the MC circuits connecting the dorsal MOB and CoA. In summary, it has become clear in the present study that activation of a single guidance gene, *Nrp2*, could determine a functional lineage of MCs and is sufficient to instruct circuit formation between the MOB and OC for attractive social behaviours.

## Methods

**Mice.** Tg Goofy-Cre[63] and Tg AP2ε-Cre[30] mice were obtained from Y. Yoshihara and T. Williams, respectively. The Nrp2-floxed, Sema3F-floxed, Rosa26-STOP-lacZ and Eno2-Cre mice were purchased from The Jackson Laboratory. Goofy-Cre, AP2ε-Cre and Eno2-Cre mice were crossed with the Nrp2- or Sema3F-floxed mice to generate the OSN-specific, MC-specific and total KO mice. To generate the *Nrp2-Cre* construct, the BAC clone RP24-250G22 containing the *Nrp2* promoter was modified and the *NiCre-pA-rpsL* sequence was introduced into the translation start site of *Nrp2*, using the Counter-Selection BAC Modification Kit (GENE BRIDGES). In the *Pcdh21* gene, an enhancer/promoter site was identified at ∼10 kb upstream of the coding region[64]. The 10 kb fragment of *Pcdh21* was isolated from the BAC clone and introduced into the SKA vector. The *intron-tau-lacZ-STOP-ires-tauEYFP-pA* sequence was inserted into the *Pcdh21* minigene. Mice

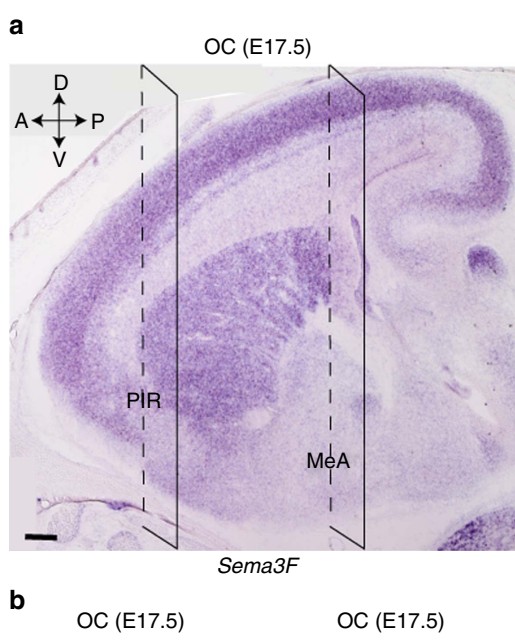

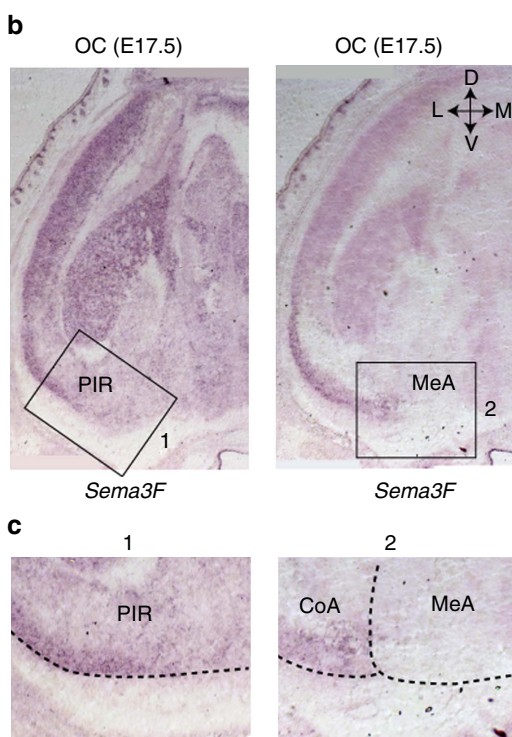

**Figure 7 | *In situ* hybridization of *Sema3F* expression in the OC.**
(**a**) A parasagittal section at E17.5 was hybridized with the *Sema3F* probes.
(**b**) Coronal OC sections containing the PIR and MeA in **a** were analysed.
*Sema3F* transcripts are high in the PIR, but rarely detected in the MeA at the
early embryonic stage. (**c**) Enlarged photos of PIR (1) and MeA (2) are
shown in **b**. Scale bar, 200 μm.

were housed in a 12-h light/dark schedule and had free access to the food and
water. Sample sizes were determined based on the standard procedure. No method
was used to randomize the animals among the experimental group. The
investigator was not blinded to the genotype, except for the pup suckling. Animal
experiments were performed in accordance with the guideline of Animal Care
Committees in the University of Tokyo, University of Fukui, Azabu University and
RIKEN Institute in Kobe.

**Two-choice preference test.** We used an acryl box (300 × 150 × 155 mm) con-
taining three compartments—one is for mice (95 × 130 mm) and two are for
samples (190 × 65 mm). Urine samples were collected for 5 consecutive days from

10 male and 10 female mice. Each sample (60 μl) was spotted on a filter paper
(40 × 40 mm) and placed in a sample compartment. The partition had a vent to
allow air to pass and mice to poke their snouts through. An air stream was drawn
through a charcoal filter from each sample compartment to the mouse compart-
ment with a vacuum pump. In the preference test, MC-specific Nrp2 cKO and
littermate male (8–12 weeks old) were analysed. Each adult mouse was habituated
for 5 min before the test. During subsequent 5 min, the time amount that a mouse
spent in poking its nose was measured (investigation time). The mouse behaviour
was recorded with a digital video camera.

**MTMT preference test.** Castrated male urine was collected from five animals.
Sexually experienced female mice were used to analyse the behavioural effects of
MTMT in a urine preference test[17]. A piece of filter paper spotted with 50 μl of
castrated male urine alone or castrated male urine with 20 ppb MTMT was
presented to the MC-specific Nrp2 cKO and littermate mice (8–12 weeks old).
Each adult mouse was habituated for 5 min before the test. In the subsequent 5-min
duration, the amount of time spent for poking the nose was measured
(investigation time). The animal behaviour was recorded with a digital video
camera.

**Recording of ultrasonic vocalization.** USV was detected using a condenser
microphone (UltraSoundGate CM16/CMPA, Avisoft Bioacoustics) that was con-
nected to an A/D converter (UltraSoundGate 116; Avisoft Bioacoustics) with a
sampling rate of 400 kHz. Acoustic signals were transmitted to a vocalization
analysis system (SASLab Pro; Avisoft Bioacoustics). Both Nrp2 MC-specific Nrp2
cKO and littermate male (8–12 weeks old) were analysed. $n = 10$ for each genotype.
Each adult male was habituated for 5 min before the test. The vocalization was
recorded for 6 min and the number of USVs was counted.

**Viral injection.** The mutant RV-ΔG-EGFP was obtained from Dr Callaway, and
amplified. RV-ΔG-EGFP-EnvA was produced as follows. BHK-EnvARGCD cells
were plated in 12-well plates at 2E5 cells per well. The following day, the glyco-
protein-deleted RV SADDG-EGFP was added at an mutiplicity of infection (MOI)
of 1.5. The following day, the cells in each well were trypsinized and replated into a
10 cm plate. Virus-containing supernatants were collected 2 days later, filter ster-
ilized and frozen at 80 °C in 1 ml aliquots[48]. The AAV vector encoding TVA, rabies
glycoprotein and mCherry (pFBAAV-CAG-mCherry-TVA-B19G), was
constructed from pAAV-EF1a-FLEX-GTB (a kind gift from Dr Callaway), followed
by viral packaging in Sf9 cells and purification as described[65]. For trans-synaptic
labelling, 0.5 μl of AAV2/1-CAG-mCherry-TVA-B19G ($2.0 \times 10^{12}$ genome
copy ml$^{-1}$) was injected into the brain at P60. During surgery, animals were
anaesthetized with 400 mg kg$^{-1}$ chloral hydrate and 44 mg kg$^{-1}$ xylazine. For
injection to the MeA, a needle was placed A/P $-1.04$ mm, M/L $-1.9$ mm from
the bregma, D/V $-5.4$ mm from the surface. For injection to the PIR, a needle was
placed A/P $+1.5$ mm, M/L $-2.5$ mm from the bregma, D/V $-4.8$ mm from the
surface. After 2 weeks, RV-ΔG-EGFP-EnvA ($10^7–10^8$ plaque forming unit ml$^{-1}$)
was injected into the same coordinate, MeA or PIR. Animals were then housed for
10 days to allow the virus to infect and spread trans-synaptically.

***In utero* electroporation.** For *in utero* electroporation, pregnant mice were
anaesthetized by intraperitoneal injection of ketamine (100 mg kg$^{-1}$) and xylazine
(10 mg kg$^{-1}$). The uterine horns were removed from the abdominal cavity[50].
Approximately 0.5 μl of DNA solution (1.5–4 μg μl$^{-1}$ in 50% TE) was injected into
the lateral cerebral ventricle of embryos. DNA solution was mixed with Fast Green
(100 μg ml$^{-1}$) to visualize the injection site. Electroporation was performed by
applying square electric pulses. Two pulses of 30 V were given for 50 ms with a
950 ms interval. Positive current was given from posterior to anterior in the MOB
to efficiently label the MC precursors. The uterine horns were repositioned in the
abdominal cavity.

**Suckling behaviour of pups.** Lactating female mice were anaesthetized and laid
on the side exposing nipples towards pups. P0 pups were fasted for 4 h before the
experiment, and placed 1 cm away from the female's nipple. The time for finding
nipples to suckle was measured within 4 min. Each pup was used only once in one
stimulus condition. Experiments were performed in blind to the genotype of pups.
The genotype was examined after each experiment.

**Fear responses to TMT.** Mice were habituated to a clean cage (23 × 16 × 12 cm)
for 5 min before the test. A piece of filter paper spotted with 10 μl of undiluted
TMT (Contech Enterprise Inc.) was presented to mice for 10 min. MC-specific
Nrp2 cKO and littermate mice (8–12 weeks old) were analysed. $n = 10$ for each
genotype. The mouse behaviour was recorded with a digital video camera. Average
freezing time was measured.

**Habituation–dishabituation test.** A small perforated tube (1.5 ml microtube) with
a piece of cotton spotted with 10 μl of 1% vanillin (or undiluted α-terpene) was
presented to the mouse for 1 min. This was repeated four times with 10-min

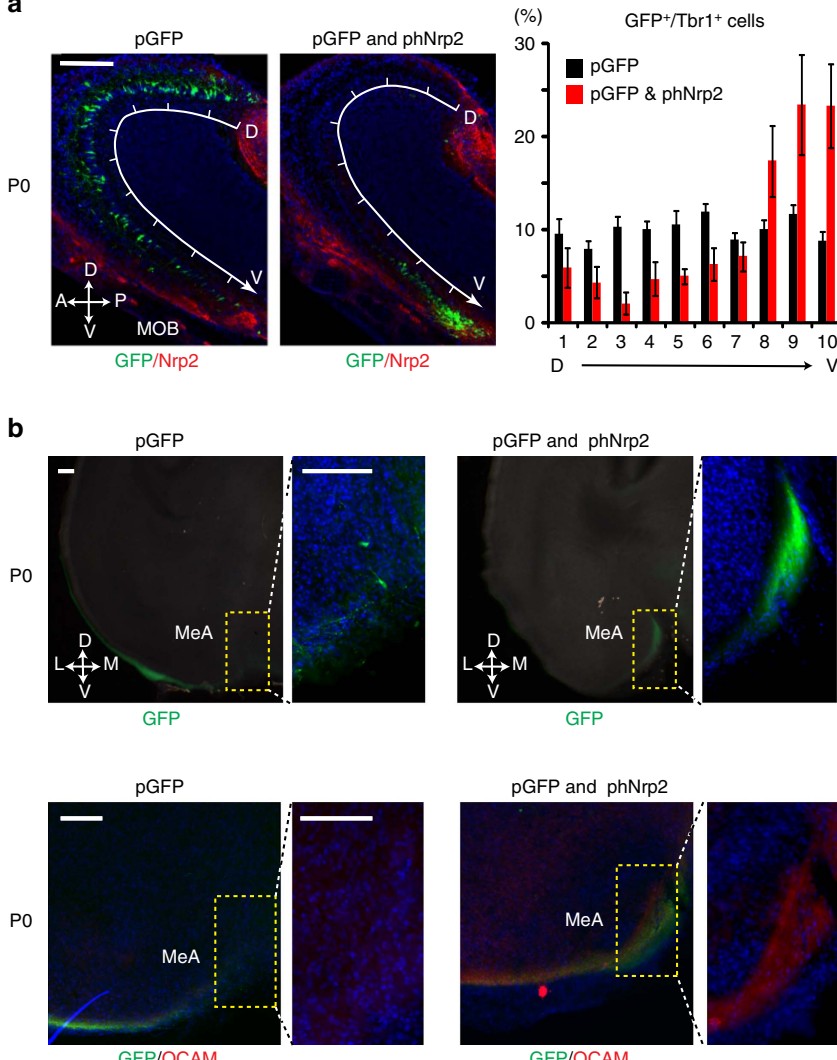

**Figure 8 | *In utero* electroporation of the *Nrp2* gene.** (**a**) *In utero* electroporation of the embryonic MOB. Plasmid vectors containing *EGFP* cDNA (pGFP) with or without human *Nrp2* cDNA (phNrp2) were introduced into the MOB at E11. Parasagittal MOB sections at P0 were immunostained with antibodies against GFP (to detect ectopic hNrp2$^+$ MCs) and mouse Nrp2 (to detect endogenous Nrp2$^+$ MCs) (left). Note that the anti-mouse Nrp2 antibodies do not detect the human Nrp2. The MOB was dissected into 10 sections along the D–V axis and the distribution of GFP$^+$ Tbr1$^+$ MCs was analysed. Statistical data are shown in the right. Ectopic expression of Nrp2 alone appears to direct migration of EGFP$^+$ MCs to the PV MOB. Data are presented in mean ± s.e. Scale bar, 200 μm. (**b**) Projection of electroporated hNrp2$^+$ MCs to the MeA. Coronal sections of the OC at P0 were immunostained with anti-GFP (green) and anti-OCAM antibodies (red). Note that the dorsal MC marker (OCAM) is detected with EGFP in the MeA, when the hNrp2 is ectopically expressed in MCs. Scale bar, 100 μm. Nuclei are counterstained with DAPI in blue (**a**,**b**).

intervals. In the fifth trial 10 min later, a 10 μl drop of undiluted α-terpinene (or 50 mM, 0.5 μM vanillin) was placed in the mouse cage. Habituation is defined by the progressive decrease of sniffing towards the repeated presentation of the same odour. Dishabituation is refined by the reinstatement of sniffing when the novel odour is presented. MC-specific cKO of Nrp2 and littermate mice (8–12 weeks old) were analysed. $n = 6$ and 4 for each genotype in the 50 mM and 0.5 μM vanillin experiments, respectively.

***In situ* hybridization.** After perfusion with 4% paraformaldehyde (PFA) in phosphate-buffered saline (PBS), the brain samples were taken out from mice and fixed overnight with 4% PFA in PBS. Parasagittal or coronal sections (10–20 μm each) were prepared from the frozen tissues in OCT compound (Tissue-Tek). To generate RNA probes, DNA fragments of 500–1,000 bp were amplified from cDNA of the mouse brain (C57BL/6). PCR products were subcloned into pGEM-T (Promega) and used as templates for RNA probes. To label the probes with digoxigenin (DIG), the DIG RNA labelling kit (Roche) was used. For *in situ* hybridization, samples were fixed in 4% PFA for 15 min at 4 °C (refs 11,66). After rinsing with PBS, the sections were incubated with 7 μg ml$^{-1}$ proteinase K for 10 min at 37 °C. The samples were fixed again with 4% PFA in PBS for 10 min, incubated with 0.25% acetic anhydride and 0.1 M triethanolamine, pH 8.0 and

washed with PBS. For hybridization, the sections were incubated for 16 h at 51 °C with probes. After washing, the samples were blocked with the blocking reagent (Roche), and incubated with the alkaline phosphatase (AP)-conjugated anti-DIG antibody (Roche). Probe-positive cells were stained purple with nitroblue tetrazolium salt and 5-bromo-4-chloro-3-indolyl phosphate toludinium salt. For fluorescence staining, probe-positive cells were incubated with a HNPP/FastRed (Roche) in the staining buffer (100 mM Tris-Cl pH 8.0, 100 mM NaCl, 5 mM MgCl$_2$). After washing, samples were photographed with a confocal microscope, model BX61 (Olympus). Boundaries of brain structures were determined according to the standard atlas[67,68].

**Immunohistochemistry.** Immunohistochemistry was performed according to the published procedure[12] using the following antibodies: anti-Nrp1 (1:500; R&D systems), anti-Nrp2 (1:500; R&D systems), anti-GFP (1:1,000; Clontech), anti-β-galactosidase (1:1,000; ICN Biochemicals), anti-OCAM (1:500; R&D systems), rabbit anti-Tbr1 (1:5,000; Abcam), rabbit anti-Tbr2 (1:5,000; Abcam), anti-BrdU (1:200; Abcam) and rabbit anti-Tbx21 (1:10,000; provided by Dr Yoshihara at RIKEN). Antibodies against Pcdh21 were generated by immunizing guinea pigs with KLH-conjugated synthetic peptides as described previously[11]. Alexa fluor-conjugated secondary antibodies (Invitrogen) were used at 1:200 dilution.

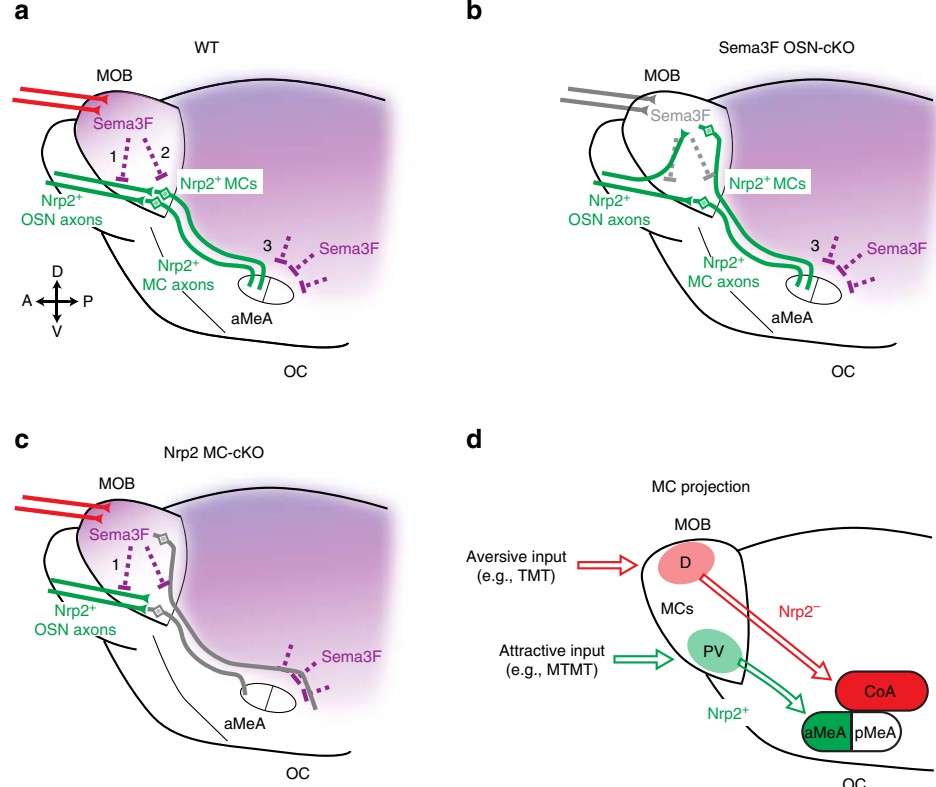

**Figure 9 | Schematic diagrams of Nrp2$^+$ MC targeting. (a)** Sema3F–Nrp2 interactions in the olfactory circuit formation in the WT. A repulsive ligand, Sema3F, expressed by the dorsal-region OSNs is essential for guiding both Nrp2$^+$ OSN axons (1) and *Nrp2$^+$* MCs to the ventral OB (2). OC-region Sema3F instructs axon targeting of *Nrp2$^+$* MCs to the MeA (3). **(b)** Targeting of Nrp2$^+$ MCs in the OSN-specific Sema3F cKO. Because of the absence of Sema3F in the dorsal MOB (1), distribution of Nrp2$^+$ MCs (2) is perturbed. However, targeting of Nrp2$^+$ MCs (3) to the anterior MeA (aMeA) is not affected. Nrp2$^+$ MCs remaining in the dorsal MOB in the cKO send their axons to the aMeA. **(c)** Targeting of MCs in the MC-specific Nrp2 cKO. Because of the absence of Nrp2 in MCs, both distribution (2) and targeting (3) of MCs are impaired in the cKO. MC axons from the MOB are significantly reduced in the aMeA. **(d)** A schematic diagram of the secondary projection in the mouse olfactory system. Aversive signals are reported to be transmitted from the dorsal MOB to the CoA[14,15]. Nrp2$^+$ MCs convey the attractive social signals from the PV MOB to the aMeA but not to the posterior region (pMeA). Our present study revealed that activation of a single axon guidance gene, *Nrp2*, in MCs is sufficient to instruct olfactory circuit formation from the MOB to MeA.

**X-gal staining.** Olfactory tissues were prepared after perfusion with 4% PFA in PBS. The sections were washed with PBS and placed in X-gal staining buffer (5 mM potassium ferricyanide, 5 mM potassium ferrocyanide, 2 mM MgCl$_2$, 0.01% sodium deoxycholate, 0.02% Nonidet P-40 (NP-40), 1 mg ml$^{-1}$ X-gal in PBS). The samples were incubated overnight at 37 °C in the dark. Photographs were taken with an Olympus Optical AX70 microscope (Olympus).

**BrdU labelling.** Labelling of 5-bromodeozyuridine (BrdU) was performed as described previously[69]. BrdU was intraperitoneally injected into the pregnant mice (100 mg kg$^{-1}$) at pregnant day 9, 10, 11, 12, 13 and 14. OB sections were prepared from the E18.5 embryo, and washed three times with PBS at room temperature for 5 min, incubated in 1.2 N HCl at 37 °C for 1 h, rinsed in 100 mM borate buffer (pH 8.5) at room temperature for 1 min twice and in PBS briefly. Then, the BrdU was stained with anti-BrdU antibodies.

**Plasmids.** A plasmid vector containing the *egfp* cDNA with CAG enhancer (pGFP) was obtained from Addgene (pCAG-GFP; Plasmid #11150). To construct pNrp2, full-length human nrp2 cDNA was cloned with PCR using the template plasmid (Clone; HsCD00443185) from DNASU. Primer sequences used were: 5′-AGTGATATCCACCATGGATATGTTTCCTCTCACC-3′; and 5′-AGTGA-TATCTCATGCCTCGGAGCAGCACTTTTGG-3′. The obtained PCR fragment was inserted into the pCAGEN vector (Addgene; Plasmid #11160).

**Intensity measurements and statistical analyses.** Optical images were photographed with an Olympus Optical AX70 microscope. Sections were also analysed by a fluorescence microscope, Olympus model IX70 equipped with a cooled CCD camera, C4742-95-12ERG (Hamamatsu Photonics). For quantification of signals, the tone was reversed and monochrome images were used. Staining intensities were measured with ImageJ. Statistical analyses were performed with Excel 2010 (Microsoft).

**Data availability.** The data that support the findings in this study are available from the corresponding author on request.

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

## Acknowledgements

We thank Y. Yoshihara for Goofy-Cre mouse, T. Williams for AP2ε-Cre mouse, E. Callaway for RV and TVA vectors, E. Block and H. Matsunami for MTMT. We are grateful to

L. Luo, A.J. Otsuka and members of our laboratory for valuable comments and discussion. K.I. was a pre-doctoral fellow of the Japan Society for the Promotion of Science, and H.N. was supported by Urakami Foundation. This work was supported by Grants in Aid from the Ministry of Education, Culture, Sports, Science and Technology (MEXT) of Japan (T.K., H.N., H.T., H.B. and H.N.), and by the JST-PRESTO (H.T.) and AMED-CREST (H.B.) programmes. H.S. is funded by the Specially Promoted Research Grants from MEXT of Japan, and F.I. by the NIH Grant, RO3-DC011134. All recombinant DNA and animal experiments were performed in accordance with the regulations and guidelines of the Universities, and approved by the institutional review committees of Graduate School of Medicine and Graduate School of Science, University of Tokyo; School of Medical Science, University of Fukui; and School of Veterinary Medicine, Azabu University.

## Author contributions

This research was conceived by K.I., H.T., H.N. and H.S. K.I. performed most of the experiments. *In utero* electroporation experiments were done by F.I. H.O. purified AAV and RV. K.I. and R.K. performed virus injection, and H.B. supervised the virus studies. T.K. oversaw the behavioural analysis. The manuscript was prepared by K.I., H.T., H.N. and H.S.

## Additional information

**Competing interests:** The authors declare no competing financial interests.

