## [Peer Review File · Nature Communications]

Reviewers' comments:

Reviewer #1 (expertise in olfactory circuits) (Remarks to the Author)

The authors show that:

Nrp2+ MCs in PV are OB important for a number of social attraction cues:

- MC NrP2 KO in males reduces investigation time of female urine compared to male urine.
- MC NrP2 KO females investigated castrated male urine NSD from castrated male urine plus MTMT.
- USV signals in MC NrP2 KO males in response to female intruders were significantly reduced compared to controls.
- MC Nrp2 cKO pups had significant deficits in locating nipples of a anaesthetised doe compared to controls. This was not due to locomotion deficits because reflex righting was unimpaired.

Nrp2 and Sema3F interactions are important for correct targeting of OSNs and MCs in the OB and for MCs to the MeA:

- KO of Nrp2 in OSNs prevents them becoming localised in the PV OB, but does not affect MC expression patterns.
 - KO of Nrp2 in MCs prevents MCs becoming localised in PV, but does not alter OSN expression patterns.
- This suggests PV localisation of OSNs and MCs is not due to Nrp2 interactions between OSNs and MCs.
- KO of Sema3F in OSNs impairs localisation of both Nrp2+ MCs and OSNs.
 - EYFP expression in Nrp2+ MCs showed that they project their axons to the MeA, while dorsal OCAM expressing MCs do not.
 - Transsynaptic retrograde rabies labelling shows that ventral, but not dorsal MCs project their axons to the MeA.
 - Sema 3F KO shows reduced projection of MCs to MeA.
 - Ectopic expression of hNrp2 is enough to make OB cells accumulate in PV and can even induce dorsal MCs to migrate to the MeA.

The data largely seems to support the authors' claims. However, to my mind there are some improvements that could be made that would substantially improve the paper

Fig 1a: Without references, this is a hypothetical diagram. If relevant references exist then they should be cited .. particularly to back up claim PV neurons do not project to pMeA. It may also be helpful for the reader if colouring throughout the document is consistent. I.e. colours for PV projections in fig. 1a do not match Nrp2 in rest of figure.

Fig 1c: I would benefit from some discussion as to why Nrp2 cKO female responses to castrated male mice is so much larger than both cast+MTMT and WT response to cast mice.

Fig S1b/Lines 113-117: Unsurprising that the habituation/dishabituation test showed that detection of some chemical odours were not affected by cKO of PV nrp2 because terpin-4-ol (McBride Slotnick 2006) and vanillin (Oka 2006) activate dorsal glomeruli. Perhaps discuss.

Fig S2c: Figures don't appear to be in the order they appear in the text. Fig S2c should be put first and renamed S2a. No evidence that Nrp2- are segregated from Nrp2+ cells in MOB is provided here. The Hinds (36) and Blanchart (37) references quoted make no reference to Nrp2 expression. It is well possible that there are Nrp2- MCs in the PV region mingled with the Nrp2+ neurons. Fig 2a also clearly shows posterodorsal expression of Nrp2+ cells so expression of Nrp2+ neurons is

not restricted to PV areas.

Figure 2a/Line 150: The authors should provide evidence as claimed that Nrp2 is only expressed during embryonic stages. Fig 2a shows P0 MOB - i.e. no longer embryonic, so why is there still Nrp2 expression? Some discussion as to how posterodorsal Nrp2 expression may affect results seems relevant

Lines 150-151. Perhaps clarify the use of term "nrp2-". Is this intended to be the same as mature pcdh21 neurons, some of which could presumably have nrp2+ history? Presumably there are also immature nrp2- ap2E+ cells expressed ventrally?

Fig S2a/Line 153: Dependent on alteration of line 150-151, perhaps change "Nrp2-" to "pcdh21+" or "mature MCs". Nrp2+ cells represent a subset of MCs in that area. Assuming I am correct then some change in text to reflect this seems appropriate.

Line 154. There is clearly some overlap of Nrp2+ and pcdh21 cells. Fig S2b confirms that there are nrp2+ Pcdh21 cells. So pcdh21 is not a pure marker of Nrp2- cells. Discuss or change text to reflect this.

Lines 156-157: Fig 2a shows Nrp2+ MCs are not confined to PV region but also PD. Change text to reflect this. Fig 2b: No evidence that all PV OSNs are nrp2-. Alter text to reflect this.

Line 158/9: No direct evidence that nrp2- OSNs don't connect with nrp2+ OSNs. Please include this in the discussion.

Fig S4a/Lines 199 - 200 : Please discuss shell of B-gal fluorescence around the AOB.

Fig S4b/Lines 204-205: Please provide a bigger section showing lack of EYFP signal absence in dorsal MCL.

Fig 4b/Line 238-239 shows there are clearly some projections from dorsal region to MeA. Please alter text to reflect this.

Figure 6: Perhaps put relevant data from the supplementary figures into this figure.

Fig 6c: Colouring of cartoon suggests Sema 3F is weakened, however, it's the MC responses that are weakened isn't it?

Line 276 - Perhaps add that Tbr2 and Tbx21 were also expressed into main text. Not all transfected cells were necessarily MCs - perhaps this also needs to be discussed.

Fig 7b: Presumably nrp2 cKO mice were used for these experiments? If so, please clarify in the text. If they are WT, please discuss why is there not normal expression of eGFP in MeA as there is in figure 3b.

Discussion:

Based on recommendations above I felt there needs to be substantial alternations and lengthening of the discussion..

Please change lines 341-342 to:

It is probable that the MOB developed a subset of Nrp2+ MCs that allowed axons to project to the MeA using the Nrp2/Sema3F guidance system, similar to the AOB. This may have conferred the

evolutionary advantage of detecting volatile social cues from distances.

Materials and Methods:

Few of the behavioural experiments are described in enough detail to allow replication. The authors should include all parameters for USVs, investigation times, suckling, righting and freezing should be described or referenced. How were MTMT experiments performed? What concentration of MTMT was used? Include whether the experimenter was blind to which mice were mutants. Age of pups and please ensure age of pups was NSD in each treatment group.

The authors need to explain how boundaries of brain structures were determined.

Reviewer #2 (expertise in olfactory circuits, physiology)(Remarks to the Author):

In this manuscript Sakano et al describe the role of Neuropilin-2 (Nrp2) expressing mitral cells in regulating specific social attraction behaviors. By deletion experiments they demonstrate that Nrp2 is necessary for appropriate targeting of mitral cells and social attraction behaviors. Sufficiency is shown by ectopic expression of Nrp2 resulted in targeting of mitral cells to these specific circuits.

Generally this is a well designed and interesting set of experiments addressing an important question about the structure and functional subdivisions of the olfactory bulb. The approaches used for generating specific alterations in expression of Nrp2 and Sema3F in specific neuronal populations are elegant and convincing. The role of Nrp2 in specifying these circuits for social attraction is novel and interesting and the mechanistic aspects of this study are also surprising. This work provides further evidence of the existence of specific olfactory subsystems that mediate specific behavioral repertoires.

Concerns:

The interpretation of the Nrp2 KO is not clear. DO the authors believe that by deleting Nrp2 they have eliminated a class of mitral cells or (I assume) that they have simply eliminated Nrp2 expression from a specific class of mitral cells? This interpretation should be clarified.

The paper would be strengthened if there was additional analysis of the projections of the Nrp2 MCs to the piriform cortex. The specificity of projections by mitral cells to areas such as the amygdala may co-exist with more generalized projections to other olfactory bulb target areas, as has been suggested by previous anatomical analyses. Perhaps extensive analysis is outside the scope of this paper, but some comment on this issue would be appropriate.

Authors claim that the Nrp2 KO does not alter detection of chemical cues (line 113). This is an overly broad generalization, as it certainly may be that detection of some chemical cues, including those involved in the behaviors described are altered. This statement should be clarified.

Line 158 The statement that mitral cells send their dendrites “to the nearest glomeruli” is an over simplification and should be clarified. The pattern of dendritic innervation and the biases have been described (mitral cell dendrites innervate glomeruli anterior to their cell body location) but not extensively characterized for the ventral regions of the bulb.

Reviewer #3 (expertise in olfactory circuits) (Remarks to the Author):

Inokuchi et al., perform a nice series of experiment to investigate the developmental program that specifies connections from the olfactory bulb to higher brain centers. It had been suggested that the ventral olfactory bulb mediates innate social behaviors via projections to the medial amygdala (MEA). Here, the authors directly test this hypothesis by knocking out the Nrp2 receptor from mitral cells (MC), which disrupts MC axon targeting to MEA as well as odor induced social behavior. Moreover, ectopic expression of Nrp2 in dorsal MCs is sufficient to convert them to ventral MCs with projections to MEA. The authors further demonstrate that the axon guidance molecule, *Sema3*, is also required for the proper targeting of MC axons to MEA, presumably through Nrp2 signaling. Overall, this is a very nice set of experiments that address an important question and is appropriate for publication in Nature Communications with minor revisions. I have the following minor concerns.

1) Figure 1a is presented as a diagram of the mouse olfactory system. It should be clearly stated that this diagram is a model, rather than fact. In particular, the Ventral bulb to MEA has not been firmly established prior to this paper.

2) In Figure 1b, the authors examine male investigation of urine isolated from male vs. female mice. Is the female urine collected from estrous females? The authors should include this information in methods. The same comment applies to the female mice assayed for behavior in Figure 1c.

3) In Figure 1c, the Nrp2 MC-KO mice appear to have increased attraction to castrated male urine, relative to controls, despite the decrease in attraction to the MTMT odorant. Is this difference significant? If so, what is going on here?

4) Figure 1 d, shows spectrograms of ultrasonic vocalizations. These images are hard to see because the wispy white line is faint on the black background. The authors should consider inverting the color scheme and/or increasing the line thickness.

5) Figure 1f lacks test of significance and error bars. A simple test for proportions, such as the Z-test or Chi-square test should be applied. In addition, the right bars (“No-suckling”) are redundant with the left bars as it is logically the subtraction of “Suckling” from 100.

6) In Figure 2, the authors examine the OSN projection to the olfactory bulb and the MC positions, with various manipulations. It is unclear why the authors sample these two neuronal populations at different developmental time points. This is generally unclear throughout the paper. The authors should clearly explain their rationale for using different aged mice throughout.

7) The Figure 2 legend lacks any mention of the number of samples or biological replicates the authors examined.

8) Sup. Figure 3 seems to be entirely redundant with Figure 2.

9) In Figure 3, the authors compare the axon projections from dorsal vs ventral MCs to higher order olfactory areas. It is difficult to compare the projections because the coronal sections between the two groups do not appear to be from the same positions along the anterior-posterior axis; the sections for the dorsal MC axons seem to be consistently more anterior than the ones for ventral projections. The same critique applies to Sup. Figure 4e. Further, in the right most image (Sup. Fig 4e), what is labeled as CoA, looks like the nucleus of the lateral olfactory tract, which is more anterior to the CoA shown in the left image. Again, why are the authors showing these groups from different developmental stages? There is also no indication of the number of samples examined.

10) In Figure 5, the authors compare the ventral projections to the MEA and use Nrp1, rather than Nrp2, as a marker for these Nrp2+ MCs. They do this because they delete the Nrp2 gene and need another marker for these neurons. However, they do not clearly establish that Nrp1 is expressed in the Nrp2+ MCs. The authors should show this at the olfactory bulb level where single cell resolution is possible. The broad colocalization of axons (Sup Figure 4c) is inadequate to address the colocalization of Nrp1 and Nrp2 in ventral MC.

11) In Figure 7, the authors perform an elegant experiment in which they ask ectopic expression of Nrp2 is sufficient to instruct MCs to become ventral MCs projecting to MEA. This is well done. However, this figure is interrupted by a rather negligible experiment in which they ask if the neurons that ectopically express Nrp2 also express Tbr1, another marker for MCs. They claim that because MCs do express this other marker, the cells maintain MC properties. This is weak evidence to support this claim. The real question is whether these gain-of-function neurons form functional circuits by receiving OSN input and making synaptic connections to MEA. If the authors wish to pursue this, one could simply assay for odor evoked c-fos expression, to determine if the MCs respond to odor and relay activity to MEA.

12) The summary cartoon, Figure 6, would seem more appropriately placed as the last figure.

Reviewer #4 (expertise in axon guidance) (Remarks to the Author):

In this manuscript, Inokuchi and colleagues investigate the wiring of a mouse olfactory circuit involved in social odor cues, from mitral cells (MCs) in the ventral olfactory bulb (OB) to the anterior part of the medial amygdala (MeA).

They first establish that a mitral cell-specific deletion of the Nrp2 gene perturbs behaviors elicited by positive social odor cues. For example, male attraction to female urine, and ultrasonic vocalization towards females was substantially reduced in the Nrp2 KO mice. In contrast, fear responses induced by TMT, which are thought to be mediated by the dorsal OB via the cortical amygdala, were unaffected by the mutation.

The authors then determine the functions of the Nrp2-Sema3F guidance-receptor pair in establishing olfactory sensory neuron (OSN) – MC connectivity in the olfactory bulb. Using a series of cell type-specific loss of function experiments they show that Nrp2 and Sema3F deletions causes mistargeting of OSN axons and a redistribution of ventral MCs towards more dorsal locations in the OB, demonstrating that Nrp2/Sema3F signaling is required for the proper matching of connections in the ventral OB.

The authors next use both anterograde and retrograde neural tracing experiments to show that Nrp2-positive MCs from the ventral OB project to the medial amygdala (MeA). They generate two new transgenic mouse lines to specifically label Nrp2-positive MCs, and establish that Nrp2-

positive neurons project axons to several cortical targets, including the MeA. Furthermore, retrograde tracing of neurons that project to the MeA, using a glycoprotein-deleted rabies virus for trans-synaptic, reveals that MeA-projecting mitral cells are preferentially located in the ventral, Nrp2-positive OB.

Finally, the authors report that Nrp2 is necessary and sufficient for the establishment of OB-MeA connections. MC-specific deletion of Nrp2 reduced MC innervation of the MeA, while leaving other targets unperturbed. Moreover, misexpression of Nrp2 in MCs including in the dorsal OB, via in utero electroporation, results in the rerouting of dorsal MCs, defined by the expression of OCAM, towards the MeA.

Together, these results provide important new insights into how molecular determinants of cell migration and axon guidance instruct the wiring of a behaviorally relevant olfactory neural circuit. While mechanisms of axon guidance and topographic map formation have extensively been studied in the OB, our understanding of the wiring of central olfactory projections remains rudimentary. This is of particular and general relevance as the character of the olfactory map in the OB versus the different primary cortical areas is dramatically different. The identification of Nrp2 as a key determinant for establishing the connectivity of OSNs-MCs-MeA neurons thus provides exciting and important new mechanistic insights into how proper circuit wiring is achieved.

The authors use an elegant, impressive array of genetic tools. Overall, the data is convincing and the figures, including supplements, are well organized and described. Use of statistics appears appropriate. I thus strongly recommend publication of this manuscript in Nature Communications, and I only have a few minor remarks and suggestions.

Minor remarks:

Retrograde tracing:

The authors state that starter cells were confined to the MeA. However, Figure 4b only shows a high magnification image of the MeA. The authors should show a larger area including surrounding areas. (line 230)

Figure 1:

To illustrate behavioral results, the authors should show individual data points for individual mice, rather than the bar graph only.

Figure 2:

It would be preferable to include Neurotrace counterstain in all panels of Figure 2c and d.

Discussion:

The authors show that MC-specific Nrp2 deletion results in reduced innervation of the MeA. It appears possible that this effect is caused by a reduction of MeA-projecting MCs, a reduction in the formation of MeA collaterals in these cells, or by a rerouting of axons to other targets. The authors should discuss these different possibilities.

More generally, it would be interesting if the authors could expand their discussion on the formation of cortical olfactory maps. Sparse labeling of MCs suggests that MCs form collaterals that project to multiple cortical targets (Ghosh et al., 2011; Igarashi et al., 2012). Strikingly, these different cortical areas appear to exhibit very different degrees of topographic organization. Thus, individual MCs may rely on the graded expression of classical guidance factors for establishing target specificity of one collateral, but use possibly different strategies to establish non-topographic connections of another collateral. While the authors briefly refer to the timing of Nrp2

and Sema3F expression as a possible explanation, it would be helpful to discuss this point in more detail.

References:

References 21 and 21 do not appear to describe responses of the MeA to attractive odor cues, as indicated in the introduction. (line 64)

Point-by-point responses to each reviewer are as follows (reviewer's comments are in blue italics):

Reviewer #1 (expertise in olfactory circuits) (Remarks to the Author)

The authors show that:

Nrp2+ MCs in PV are OB important for a number of social attraction cues:

-MC Nrp2 KO in males reduces investigation time of female urine compared to male urine.

-MC Nrp2 KO females investigated castrated male urine NSD from castrated male urine plus MTMT.

-USV signals in MC Nrp2 KO males in response to female intruders were significantly reduced compared to controls.

-MC Nrp2 cKO pups had significant deficits in locating nipples of an anaesthetised doe compared to controls. This was not due to locomotion deficits because reflex righting was unimpaired.

Nrp2 and Sema3F interactions are important for correct targeting of OSNs and MCs in the OB and for MCs to the MeA:

-KO of Nrp2 in OSNs prevents them becoming localized in the PV OB, but does not affect MC expression patterns.

-KO of Nrp2 in MCs prevents MCs becoming localized in PV, but does not alter OSN expression patterns.

This suggests PV localization of OSNs and MCs is not due to Nrp2 interactions between OSNs and MCs.

-KO of Sema3F in OSNs impairs localization of both Nrp2+ MCs and OSNs.

-EYFP expression in Nrp2+ MCs showed that they project their axons to the MeA, while dorsal OCAM expressing MCs do not.

-Transsynaptic retrograde rabies labelling shows that ventral, but not dorsal MCs project their axons to the MeA.

-Sema 3F KO shows reduced projection of MCs to MeA.

-Ectopic expression of hNrp2 is enough to make OB cells accumulate in PV and can even induce dorsal MCs to migrate to the MeA.

The data largely seems to support the authors' claims. However, to my mind there are some improvements that could be made that would substantially improve the paper

#1-1.

Fig 1a: Without references, this is a hypothetical diagram. If relevant references exist then

they should be cited .. particularly to back up claim PV neurons do not project to pMeA. It may also be helpful for the reader if colouring throughout the document is consistent. I.e. colours for PV projections in fig. 1a do not match Nrp2 in rest of figure.

As pointed out, this is a hypothetical diagram. Targeting from the ventral bulb to MEA has not been firmly established prior to our paper. To avoid confusion, we moved this schematic diagram to the end of the paper as a summary diagram in Fig. 7d. As for the coloring of projection pathways, we made necessary changes to make it consistent throughout the paper.

#1-2.

Fig 1c: I would benefit from some discussion as to why Nrp2 cKO female responses to castrated male mice is so much larger than both cast+MTMT and WT response to cast mice.

This is an interesting point. To be honest, we do not have a good explanation for it. However, we added some sentences discussing it in page 5, lines 108-110.

#1-3.

Fig S1b/Lines 113-117: Unsurprising that the habituation/dishabituation test showed that detection of some chemical odours were not affected by cKO of PV nrp2 because terpin-4-ol (McBride Slotnick 2006) and vanillin (Oka 2006) activate dorsal glomeruli. Perhaps discuss.

We agree with the reviewer. We modified sentences in lines 138-142.

#1-4.

Fig S2c: Figures don't appear to be in the order they appear in the text. Fig S2c should be put first and renamed S2a. No evidence that Nrp2⁻ are segregated from Nrp2⁺ cells in MOB is provided here. The Hinds (36) and Blanchart (37) references quoted make no reference to Nrp2 expression. It is well possible that there are Nrp2⁻ MCs in the PV region mingled with the Nrp2⁺ neurons. Fig 2a also clearly shows posterodorsal expression of Nrp2⁺ cells so expression of Nrp2⁺ neurons is not restricted to PV areas.

According to the reviewer's suggestion, we moved Supplementary Fig. 2c to 2a. As pointed out, we cannot exclude the possibility that Nrp2⁻ MCs are mingled with Nrp2⁺ cells in the PV region. We, therefore, modified the figure accordingly. As for the

Nrp2 signals in the PD region, they represent Nrp2⁺ MCs in the AOB. To avoid this confusion, we modified the figure and mentioned this fact in the legend.

#1-5.

Figure 2a/Line 150: The authors should provide evidence as claimed that Nrp2 is only expressed during embryonic stages. Fig 2a shows P0 MOB - i.e. no longer embryonic, so why is there still Nrp2 expression? Some discussion as to how posterodorsal Nrp2 expression may affect results seems relevant

As pointed out, Nrp2 expression is not totally terminated before birth. In our experiment, Nrp2 gradually decreased after birth and disappeared by PD14. This is now mentioned in page 7, lines 155-156 and the irrelevant sentence was removed. It should be noted that Nrg2⁺ signals in the PD region are from the AOB.

#1-6.

Lines 150-151. Perhaps clarify the use of term "nrp2-". Is this intended to be the same as mature pcdh21 neurons, some of which could presumably have nrp2+ history? Presumably there are also immature nrp2- ap2E+ cells expressed ventrally?

The reviewer probably assumes that Nrp2 expression is terminated when MCs become matured and thus, the Nrp2⁻ phenotype represents mature MCs. This is not correct. There are two distinct MC lineages, Nrp2⁺ and Nrp2⁻, and Pcdh21 and Ap2ε are used as developmental markers for mature and immature MCs, respectively. To clarify this point, we added some sentences in the text, lines 158-159.

#1-7.

Fig S2a/Line 153: Dependent on alteration of line 150-151, perhaps change "Nrp2-" to "pcdh21+" or "mature MCs". Nrp2+ cells represent a subset of MCs in that area. Assuming I am correct then some change in text to reflect this seems appropriate.

Here again, Nrp2⁻ does not mean mature or Pcdh21⁺ MCs. There are two subsets of mature (Pcdh21⁺) MCs, one is Nrp2⁺ and the other is Nrp2⁻.

#1-8.

Line 154. There is clearly some overlap of Nrp2+ and pcdh21 cells. Fig S2b confirms that there are nrp2+ Pcdh21 cells. So pcdh21 is not a pure marker of Nrp2-

cells. Discuss or change text to reflect this.

Nrp2⁺ and Nrp2⁻ cells can be found in both immature (Ap2ε⁺) and mature (Pcdh21⁺) MCs, but Nrp2⁺ cells are localized in the ventral IZ.

#1-9.

Lines 156-157: Fig 2a shows Nrp2+ MCs are not confined to PV region but also PD. Change text to reflect this. Fig 2b: No evidence that all PV OSNs are nrp2-. Alter text to reflect this.

Nrp2 signals seen in the PD region in the figure are for the AOB MCs. We mentioned this fact in the legend to Fig. 2a, making it clear. For the second point, “No evidence that all PV OSNs are nrp2⁻”, we have previously reported that dorsal OSN axons are Nrp2⁻, whereas PV OSN axons are Nrp2⁺ (Takeuchi et al., *Cell*, **141**, 1056-1067, 2010). This is cited in lines 167-168.

#1-10.

Line 158/9: No direct evidence that nrp2- OSNs don't connect with nrp2+ OSNs. Please include this in the discussion.

As pointed out, we cannot exclude the possibility that Nrp2⁻ MCs form synapses with Nrp2⁺ OSN axons. We, therefore, added some sentences discussing this point in lines 160-164.

#1-11.

Fig S4a/Lines 199 - 200 : Please discuss shell of B-gal fluorescence around the AOB.

We assume that these β-gal signals are due to the axons of MOB MCs passing by the AOB. This is now mentioned in lines 215-216.

#1-12.

Fig S4b/Lines 204-205: Please provide a bigger section showing lack of EYFP signal absence in dorsal MCL.

As advised, we added new photos of bigger sections to show that EYFP signals are absent in the dorsal MCL.

#1-13.

Fig 4b/Line 238-239 shows there are clearly some projections from dorsal region to MeA. Please alter text to reflect this.

We assume that these dorsal signals are probably due to diffused virus injection for tracing. However, to be precise, we modified the sentence in lines 252-254 and lines 257-259.

#1-14.

Figure 6: Perhaps put relevant data from the supplementary figures into this figure.

Since Fig. 6 is a summary diagram of the present study, we moved it to the end of the paper in Fig. 7. Related experimental data are already shown in Fig. 2 and Fig. 5.

#1-15.

Fig 6c: Colouring of cartoon suggests Sema 3F is weakened, however, it's the MC responses that are weakened isn't it?

Yes, the MC responses are weakened as pointed out. Since the coloring in Fig. 6c (Fig. 7c in the revised version) is misleading, we made necessary changes.

#1-16.

Line 276 - Perhaps add that Tbr2 and Tbx21 were also expressed into main text. Not all transfected cells were necessarily MCs – perhaps this also needs to be discussed.

We analyzed the co-expression of GFP with Tbr2 and Tbx21 in Fig. S8. As advised, this is now described in lines 296-297. In our experiment (Supplementary Fig. 8), more than 80% of EGFP⁺ cells were MCs (Imamura et al., *Mol. Cell. Neurosci.*, 54, 58-70,2013).

#1-17.

Fig 7b: Presumably nrp2 cKO mice were used for these experiments? If so, please clarify in the text. If they are WT, please discuss why is there not normal expression of eGFP in MeA as there is in figure 3b.

We used the WT, where no background GFP was detectable for Nrp2 expression. In our *in utero* electroporation experiment, at most 10% of total MCs expressed the

exogenous hNrp2 with EGFP even in the best transfection-efficiency experiment. Thus, EGFP signals in Fig.7b must be much weaker than those in Fig. 3b. This is now mentioned in lines 290-291.

#1-18.

Discussion:

Based on recommendations above I felt there needs to be substantial alternations and lengthening of the discussion..

Please change lines 341-342 to:

It is probable that the MOB developed a subset of Nrp2+ MCs that allowed axons to project to the MeA using the Nrp2/Sema3F guidance system, similar to the AOB. This may have conferred the evolutionary advantage of detecting volatile social cues from distances.

As advised, we replaced the sentences in lines 379-382.

#1-19.

Materials and Methods:

Few of the behavioural experiments are described in enough detail to allow replication. The authors should include all parameters for USVs, investigation times, suckling, righting and freezing should be described or referenced. How were MTMT experiments performed? What concentration of MTMT was used? Include whether the experimenter was blind to which mice were mutants. Age of pups and please ensure age of pups was NSD in each treatment group.

The authors need to explain how boundaries of brain structures were determined.

These experimental procedures are now described in the Methods section, lines 430-463 and lines 493-505. It is also mentioned how boundaries of brain structures were determined in the histological experiments.

Reviewer #2 (expertise in olfactory circuits, physiology) (Remarks to the Author):

In this manuscript Sakano et al describe the role of Neuropilin-2 (Nrp2) expressing mitral cells in regulating specific social attraction behaviors. By deletion experiments they demonstrate that Nrp2 is necessary for appropriate targeting of mitral cells and social attraction behaviors. Sufficiency is shown by ectopic expression of Nrp2 resulted in targeting of mitral cells to these specific circuits.

Generally this is a well designed and interesting set of experiments addressing an important question about the structure and functional subdivisions of the olfactory bulb. The approaches used for generating specific alterations in expression of Nrp2 and Sema3F in specific neuronal populations are elegant and convincing. The role of Nrp2 in specifying these circuits for social attraction is novel and interesting and the mechanistic aspects of this study are also surprising. This work provides further evidence of the existence of specific olfactory subsystems that mediate specific behavioral repertoires.

Concerns:

#2-1.

The interpretation of the Nrp2 KO is not clear. DO the authors believe that by deleting Nrp2 they have eliminated a class of mitral cells or (I assume) that they have simply eliminated Nrp2 expression from a specific class of mitral cells? This interpretation should be clarified.

As pointed out, we assume that Nrp2 KO eliminates Nrp2 expression but not a specific set of cells to be expressing Nrp2. Since it was not clear in the text how the MC-specific cKO of Nrp2 was generated, we added some sentences to describe the procedure in lines 94-98 and 412-415, as well as the related data in Supplementary Fig. 1a.

#2-2.

The paper would be strengthened if there was additional analysis of the projections of the Nrp2 MCs to the piriform cortex. The specificity of projections by mitral cells to areas such as the amygdala may co-exist with more generalized projections to other olfactory bulb target areas, as has been suggested by previous anatomical analyses. Perhaps extensive analysis is outside the scope of this paper, but some comment on this issue would be appropriate.

It has been reported that the D-V topography in the OB is roughly correlated in the AONpE but not in the PIR. Furthermore, incoming signals in the CoA are mostly from the dorsal OB. Based on these observations, we closely looked at the secondary projections of Nrp2⁺ MCs to the AON, PIR, MeA, and CoA. As shown in Figs. 3 and 4, we found that the projection of Nrp2⁺ MCs was confined to the anterior MeA, but no apparent clustering was found in other areas mentioned above. These are now

described and discussed in lines 350-356.

#2-3.

Authors claim that the Nrp2 KO does not alter detection of chemical cues (line 113). This is an overly broad generalization, as it certainly may be that detection of some chemical cues, including those involved in the behaviors described are altered. This statement should be clarified.

We agree to the reviewer's criticism. Our claim is misleading. To be accurate, we changed the sentences in lines 138-142.

#2-4.

Line 158 The statement that mitral cells send their dendrites "to the nearest glomeruli" is an over simplification and should be clarified. The pattern of dendritic innervation and the biases have been described (mitral cell dendrites innervate glomeruli anterior to their cell body location) but not extensively characterized for the ventral regions of the bulb.

We agree that our statement is oversimplified. However, there is a report showing that MCs in the PV region extend their primary dendrites perpendicularly to the nearest glomeruli (Buonviso et al., *J Comp Neurol.*, **307**, 57-64, 1991). We, therefore, modified the sentence in lines 168-170 with the reference.

Reviewer #3 (expertise in olfactory circuits) (Remarks to the Author):

Inokuchi et al., perform a nice series of experiment to investigate the developmental program that specifies connections from the olfactory bulb to higher brain centers. It had been suggested that the ventral olfactory bulb mediates innate social behaviors via projections to the medial amygdala (MEA). Here, the authors directly test this hypothesis by knocking out the Nrp2 receptor from mitral cells (MC), which disrupts MC axon targeting to MEA as well as odor induced social behavior. Moreover, ectopic expression of Nrp2 in dorsal MCs is sufficient to convert them to ventral MCs with projections to MEA. The authors further demonstrate that the axon guidance molecule, Sema3, is also required for the proper targeting of MC axons to MEA, presumably through Nrp2 signaling. Overall, this is a very nice set of experiments that address an important question and is appropriate for publication in Nature Communications with minor revisions. I have the following minor concerns.

#3-1.

Figure 1a is presented as a diagram of the mouse olfactory system. It should be clearly stated that this diagram is a model, rather than fact. In particular, the Ventral bulb to MEA has not been firmly established prior to this paper.

As mentioned by the reviewer, the figure in Fig. 1 is a hypothetical diagram prior to our present study. To avoid confusion, we moved it to the end of the paper in Fig. 7d as a summary diagram of the present study.

#3-2.

In Figure 1b, the authors examine male investigation of urine isolated from male vs. female mice. Is the female urine collected from estrous females? The authors should include this information in methods. The same comment applies to the female mice assayed for behavior in Figure 1c.

We agree that this is an important point. In the Fig.1b, we did not select estrous females. Urine of 10 females was collected for 5 consecutive days as described in the Methods section in lines 433-434. For the MTMT experiment in Fig. 1b, we followed the published protocol described by Lin *et al.* (*Nature* **434**, 470-477, 2005). These are now mentioned in lines 445-452. Detailed experimental procedures are now described in the Methods in lines 430-452.

#3-3.

In Figure 1c, the Nrp2 MC-KO mice appear to have increased attraction to castrated male urine, relative to controls, despite the decrease in attraction to the MTMT odorant. Is this difference significant? If so, what is going on here?

Yes, the difference is significant. Although this is an interesting observation, we do not have a good explanation for it. We mentioned this in lines 108-110 to discuss this issue.

#3-4.

Figure 1 d, shows spectrograms of ultrasonic vocalizations. These images are hard to see because the wispy white line is faint on the black background. The authors should consider inverting the color scheme and/or increasing the line thickness.

As advised, we converted the color in Fig. 1d to make it easier to see the USV spectrogram.

#3-5.

Figure 1f lacks test of significance and error bars. A simple test for proportions, such as the Z-test or Chi-square test should be applied. In addition, the right bars (“No-suckling”) are redundant with the left bars as it is logically the subtraction of “Suckling” from 100.

As advised, we performed the Z-test for Fig. 1f (Fig. 1e in the revised version). Since the no-suckling data are redundant as pointed out, we removed them from the figure.

#3-6.

In Figure 2, the authors examine the OSN projection to the olfactory bulb and the MC positions, with various manipulations. It is unclear why the authors sample these two neuronal populations at different developmental time points. This is generally unclear throughout the paper. The authors should clearly explain their rationale for using different aged mice throughout.

We performed these experiments at both E18 and P0. The reason why we chose the result at E18 was that *in situ* hybridization gave clearer results at E18 than P0, probably due to the higher *Nrp2* transcription-level. In contrast, we obtained better staining at P0 in immunohistochemistry for *Nrp2* expression. We mentioned these in the text, lines 194-198, to clarify the reviewer’s concern.

#3-7.

The Figure 2 legend lacks any mention of the number of samples or biological replicates the authors examined.

As advised, we added sample numbers in the legend to Fig. 2.

#3-8.

Sup. Figure 3 seems to be entirely redundant with Figure 2.

It may be redundant. However, we would like to leave Supplementary Fig. 3 as is, because the larger views of *Nrp2* distribution may be helpful.

#3-9.

In Figure 3, the authors compare the axon projections from dorsal vs ventral MCs to higher order olfactory areas. It is difficult to compare the projections because the coronal sections between the two groups do not appear to be from the same positions along the anterior-posterior axis; the sections for the dorsal MC axons seem to be consistently more anterior than the ones for ventral projections. The same critique applies to Sup. Figure 4e. Further, in the right most image (Sup. Fig 4e), what is labeled as CoA, looks like the nucleus of the lateral olfactory tract, which is more anterior to the CoA shown in the left image. Again, why are the authors showing these groups from different developmental stages? There is also no indication of the number of samples examined.

We agree that along the A-P axis, it is difficult to compare the cortical projections as pointed out by the reviewer. We, therefore, analyzed more numbers of samples in this study, which is now mentioned in the legend to Fig. 3. To avoid the confusion between the CoA and LOT, we analyzed more posterior sections and replaced old figures with new ones in Fig. 3 right. As for the developmental stages of sample analyses, we studied cortical projection at P14, because the projection is terminated around P14. For the analysis of OCAM (a dorsal marker of MCs), we performed immunostaining at P0, because OCAM expression rapidly decreases after birth. These are now mentioned in lines 230-233.

#3-10.

In Figure 5, the authors compare the ventral projections to the MEA and use Nrp1, rather than Nrp2, as a marker for these Nrp2+ MCs. They do this because they delete the Nrp2 gene and need another marker for these neurons. However, they do not clearly establish that Nrp1 is expressed in the Nrp2+ MCs. The authors should show this at the olfactory bulb level where single cell resolution is possible. The broad colocalization of axons (Sup Figure 4c) is inadequate to address the colocalization of Nrp1 and Nrp2 in ventral MC.

As pointed out, it was not clearly established before at the single-cell level whether Nrp1 is expressed in the Nrp2⁺ MCs. To address the reviewer's criticism, we performed double ISH of OB sections and found that Nrp1 was indeed expressed in the Nrp2⁺ MCs. This is now shown in Fig. S4d and mentioned in the text in lines 234-235.

#3-11.

In Figure 7, the authors perform an elegant experiment in which they ask ectopic expression of Nrp2 is sufficient to instruct MCs to become ventral MCs projecting to MEA. This is well done. However, this figure is interrupted by a rather negligible experiment in which they ask if the neurons that ectopically express Nrp2 also express Tbr1, another marker for MCs. They claim that because MCs do express this other marker, the cells maintain MC properties. This is weak evidence to support this claim. The real question is whether these gain-of-function neurons form functional circuits by receiving OSN input and making synaptic connections to MEA. If the authors wish to pursue this, one could simply assay for odor evoked c-fos expression, to determine if the MCs respond to odor and relay activity to MEA.

We totally agree to the reviewer's point: The real question is to elucidate whether these gain-of-function neurons form functional circuits by receiving OSN input and making synaptic connections to MeA. Although we actually tried some of these experiments, we found them much more difficult than we had originally thought for the following reasons. i) In the present study, we used only WT embryos in the *in utero* electroporation experiment. If we want to use the embryos of MC-specific cKO for the gain-of-function experiment, multiple crosses of mouse lines are needed. Practically, this experiment is not easy, not impossible, though. ii) Electroporation efficiencies are not so high to obtain enough numbers of transfected (hNrp2⁺) MCs connecting to a particular glomerulus in the PV region. For these reasons, at present, we cannot clarify whether these gain-of-function neurons form functional circuits by receiving OSN input and making synaptic connections to MeA. We plan to perform these experiments in collaboration with Dr. Imamura at Penn. State Univ. after sending our cKO mice to his lab. in the US. In an effort to address the reviewer's comment, we performed a new set of experiments and obtained encouraging preliminary results. Immunostaining with a synapse marker (vGlut1) at P0 revealed that transfected (hNrp2⁺) MCs indeed form synapses with OSN axons within glomeruli. Furthermore, the transfected MCs appeared to be connected with cortical neurons within the MeA. Our observations support the idea that electroporated (hNrp2⁺) MCs are integrated into the olfactory circuit between the OB and MeA. However, synapse formation with the MeA neurons needs to be confirmed at later stages of development, because MC projection to the MeA is not completed yet at P0 and synaptic signals are still weak. As we realize the importance of the functional experiment suggested by the reviewer, we mentioned it in the Discussion in lines 336-338.

#3-12.

The summary cartoon, Figure 6, would seem more appropriately placed as the last figure.

As advised, we exchanged Fig. 6 and Fig. 7.

Reviewer #4 (expertise in axon guidance) (Remarks to the Author):

In this manuscript, Inokuchi and colleagues investigate the wiring of a mouse olfactory circuit involved in social odor cues, from mitral cells (MCs) in the ventral olfactory bulb (OB) to the anterior part of the medial amygdala (MeA).

They first establish that a mitral cell-specific deletion of the Nrp2 gene perturbs behaviors elicited by positive social odor cues. For example, male attraction to female urine, and ultrasonic vocalization towards females was substantially reduced in the Nrp2 KO mice. In contrast, fear responses induced by TMT, which are thought to be mediated by the dorsal OB via the cortical amygdala, were unaffected by the mutation.

The authors then determine the functions of the Nrp2-Sema3F guidance-receptor pair in establishing olfactory sensory neuron (OSN) – MC connectivity in the olfactory bulb. Using a series of cell type-specific loss of function experiments they show that Nrp2 and Sema3F deletions causes mistargeting of OSN axons and a redistribution of ventral MCs towards more dorsal locations in the OB, demonstrating that Nrp2/Sema3F signaling is required for the proper matching of connections in the ventral OB.

The authors next use both anterograde and retrograde neural tracing experiments to show that Nrp2-positive MCs from the ventral OB project to the medial amygdala (MeA). They generate two new transgenic mouse lines to specifically label Nrp2-positive MCs, and establish that Nrp2-positive neurons project axons to several cortical targets, including the MeA. Furthermore, retrograde tracing of neurons that project to the MeA, using a glycoprotein-deleted rabies virus for trans-synaptic, reveals that MeA-projecting mitral cells are preferentially located in the ventral, Nrp2-positive OB.

Finally, the authors report that Nrp2 is necessary and sufficient for the establishment of OB-MeA connections. MC-specific deletion of Nrp2 reduced MC innervation of the MeA, while leaving other targets unperturbed. Moreover, misexpression of Nrp2 in MCs including in the dorsal OB, via in utero electroporation, results in the rerouting of dorsal MCs, defined by the expression of OCAM, towards the MeA.

Together, these results provide important new insights into how molecular

determinants of cell migration and axon guidance instruct the wiring of a behaviorally relevant olfactory neural circuit. While mechanisms of axon guidance and topographic map formation have extensively been studied in the OB, our understanding of the wiring of central olfactory projections remains rudimentary. This is of particular and general relevance as the character of the olfactory map in the OB versus the different primary cortical areas is dramatically different. The identification of Nrp2 as a key determinant for establishing the connectivity of OSNs-MCs-MeA neurons thus provides exciting and important new mechanistic insights into how proper circuit wiring is achieved.

The authors use an elegant, impressive array of genetic tools. Overall, the data is convincing and the figures, including supplements, are well organized and described. Use of statistics appears appropriate. I thus strongly recommend publication of this manuscript in Nature Communications, and I only have a few minor remarks and suggestions.

Minor remarks:

#4-1.

Retrograde tracing: The authors state that starter cells were confined to the MeA. However, Figure 4b only shows a high magnification image of the MeA. The authors should show a larger area including surrounding areas. (line 230)

We agree that a high magnification image is not enough to confine the location of starter cells. We, therefore, added a larger view in Supplementary Fig. 5, including surrounding areas, as advised.

#4-2.

Figure 1: To illustrate behavioral results, the authors should show individual data points for individual mice, rather than the bar graph only.

As advised, we showed individual data points for individual mice in Fig. 1 (a, b, d, f), and Supplementary Fig. 1c.

#4-3.

Figure 2: It would be preferable to include Neurotrace counterstain in all panels of Figure 2c and d.

As advised, we included new counter-stained figures with DAPI in Fig. 2c and d.

#4-4.

Discussion:

The authors show that MC-specific Nrp2 deletion results in reduced innervation of the MeA. It appears possible that this effect is caused by a reduction of MeA-projecting MCs, a reduction in the formation of MeA collaterals in these cells, or by a rerouting of axons to other targets. The authors should discuss these different possibilities.

As advised, we added new sentences in lines 341-343, discussing this issue of MC projection to the MeA in the MC-specific cKO of Nrp2.

#4-5.

More generally, it would be interesting if the authors could expand their discussion on the formation of cortical olfactory maps. Sparse labeling of MCs suggests that MCs form collaterals that project to multiple cortical targets (Ghosh et al., 2011; Igarashi et al., 2012). Strikingly, these different cortical areas appear to exhibit very different degrees of topographic organization. Thus, individual MCs may rely on the graded expression of classical guidance factors for establishing target specificity of one collateral, but use possibly different strategies to establish non-topographic connections of another collateral. While the authors briefly refer to the timing of Nrp2 and Sema3F expression as a possible explanation, it would be helpful to discuss this point in more detail.

We appreciate the reviewer's constructive and helpful comments for cortical projection of MCs. Based on the reviewer's advice, we added a new paragraph discussing this issue in the Discussion in lines 357-365.

#4-6.

References:

References 20 and 21 do not appear to describe responses of the MeA to attractive odor cues, as indicated in the introduction. (line 64)

We made appropriate changes for refs. 20 and 21, as advised.

REVIEWERS' COMMENTS:

Reviewer #1 (Remarks to the Author):

The authors have done a excellent job in revising their ms. I have no additional comments.

Reviewer #2 (Remarks to the Author):

The authors have address all my major concerns.

Reviewer #3 (Remarks to the Author):

The revised manuscript is much improved. The authors adequately addressed all of my concerns, and I now find the paper to be appropriate for publication.

Reviewer #4 (Remarks to the Author):

The authors have addressed all my concerns. In my opinion, the manuscript describes an interesting, well-performed and carefully analyzed set of experiments and is ready for prompt publication in Nature Communications.